# CAMSAP2 organizes a γ-tubulin-independent microtubule nucleation centre through phase separation

Tsuyoshi Imasaki[1,2,3†], Satoshi Kikkawa[1†], Shinsuke Niwa[4*], Yumiko Saijo-Hamano[1], Hideki Shigematsu[5,6], Kazuhiro Aoyama[7,8], Kaoru Mitsuoka[8], Takahiro Shimizu[1], Mari Aoki[3], Ayako Sakamoto[3], Yuri Tomabechi[3], Naoki Sakai[5,6], Mikako Shirouzu[3], Shinya Taguchi[1], Yosuke Yamagishi[1], Tomiyoshi Setsu[1], Yoshiaki Sakihama[1], Eriko Nitta[1], Masatoshi Takeichi[9], Ryo Nitta[1,3*]

[1]Division of Structural Medicine and Anatomy, Department of Physiology and Cell Biology, Kobe University Graduate School of Medicine, Kobe, Japan; [2]JST, PRESTO, Saitama, Japan; [3]RIKEN Center for Biosystems Dynamics Research, Yokohama, Japan; [4]Frontier Research Institute for Interdisciplinary Sciences, Tohoku University, Sendai, Japan; [5]RIKEN SPring-8 Center, Hyogo, Japan; [6]Japan Synchrotron Radiation Research Institute (JASRI), Hyogo, Japan; [7]Materials and Structural Analysis, Thermo Fisher Scientific, Tokyo, Japan; [8]Research Center for Ultra-High Voltage Electron Microscopy, Osaka University, Osaka, Japan; [9]RIKEN Center for Biosystems Dynamics Research, Kobe, Japan

**\*For correspondence:**
shinsuke.niwa.c8@tohoku.ac.jp (SN);
ryonitta@med.kobe-u.ac.jp (RN)

[†]These authors contributed equally to this work

**Abstract** Microtubules are dynamic polymers consisting of αβ-tubulin heterodimers. The initial polymerization process, called microtubule nucleation, occurs spontaneously via αβ-tubulin. Since a large energy barrier prevents microtubule nucleation in cells, the γ-tubulin ring complex is recruited to the centrosome to overcome the nucleation barrier. However, a considerable number of microtubules can polymerize independently of the centrosome in various cell types. Here, we present evidence that the minus-end-binding calmodulin-regulated spectrin-associated protein 2 (CAMSAP2) serves as a strong nucleator for microtubule formation by significantly reducing the nucleation barrier. CAMSAP2 co-condensates with αβ-tubulin via a phase separation process, producing plenty of nucleation intermediates. Microtubules then radiate from the co-condensates, resulting in aster-like structure formation. CAMSAP2 localizes at the co-condensates and decorates the radiating microtubule lattices to some extent. Taken together, these in vitro findings suggest that CAMSAP2 supports microtubule nucleation and growth by organizing a nucleation centre as well as by stabilizing microtubule intermediates and growing microtubules.

## Editor's evaluation

This work investigates the mechanism by which CAMSAP2 nucleates microtubule asters in vitro, with implications for its role in the cell. The authors show the importance of CAMSAP condensates for aster formation and use electron microscopy to suggest that nucleation proceeds vs stabilization of protofilaments and sheets of tubulin.

## Introduction

Microtubules are dynamic tubular polymers that contribute to fundamental processes in cells, such as cell shape determination, chromosome segregation, cilium and flagellum formation, and molecular

**eLife digest** Cells are able to hold their shape thanks to tube-like structures called microtubules that are made of hundreds of tubulin proteins. Microtubules are responsible for maintaining the uneven distribution of molecules throughout the cell, a phenomenon known as polarity that allows cells to differentiate into different types with various roles.

A protein complex called the γ-tubulin ring complex (γ-TuRC) is necessary for microtubules to form. This protein helps bind the tubulin proteins together and stabilises microtubules. However, recent research has found that in highly polarized cells such as neurons, which have highly specialised regions, microtubules can form without γ-TuRC. Searching for the proteins that could be filling in for γ-TuRC in these cells some evidence has suggested that a group known as CAMSAPs may be involved, but it is not known how.

To characterize the role of CAMSAPs, Imasaki, Kikkawa et al. studied how one of these proteins, CAMSAP2, interacts with tubulins. To do this, they reconstituted both CAMSAP2 and tubulins using recombinant biotechnology and mixed them in solution. These experiments showed that CAMSAP2 can help form microtubules by bringing together their constituent proteins so that they can bind to each other more easily. Once microtubules start to form, CAMSAP2 continues to bind to them, stabilizing them and enabling them to grow to full size.

These results shed light on how polarity is established in cells such as neurons, muscle cells, and epithelial cells. Additionally, the ability to observe intermediate structures during microtubule formation can provide insights into the processes that these structures are involved in.

motor trafficking (*Desai and Mitchison, 1997*). α-tubulin and β-tubulin align head-to-tail to form protofilaments that associate laterally grow into the microtubules (*Alushin et al., 2014*; *Nogales et al., 1999*). The exposed β-tubulin end is called the plus-end, whereas the exposed α-tubulin end is called the minus-end. The microtubule plus-end is more dynamic than the minus-end; it dynamically alternates between growth and shrinkage. The dynamic events are regulated by many microtubule plus-end-tracking proteins (+TIPs) (*Akhmanova and Steinmetz, 2015*; *Akhmanova and Steinmetz, 2008*; *Howard and Hyman, 2003*). The minus-end is less dynamic and is responsible for determining the geometry of microtubule networks by stably anchoring at microtubule nucleation sites or the microtubule organizing centre (*Akhmanova and Steinmetz, 2019*; *Dammermann et al., 2003*).

Despite the identification of many +TIPs, only a few minus-end-binding proteins (−TIPs) have been reported, including γ-tubulin and calmodulin-regulated spectrin-associated proteins (CAMSAPs). γ-Tubulin which forms a ring complex (γ-TuRC) is located at centrosomes to nucleate microtubules and stabilize the microtubule minus-ends (*Akhmanova and Steinmetz, 2015*). γ-TuRC serves as a nucleation template for centrosomal microtubule formation by aligning αβ-tubulins to form the tubular structure of the microtubule in a process called templated nucleation (*Moritz et al., 1995*; *Roostalu and Surrey, 2017*; *Wieczorek et al., 2020*; *Zheng et al., 1995*). CAMSAPs, the protein family composed of CAMSAP1-3 in vertebrates, Patronin in *Drosophila melanogaster*, and PaTRoNin (microtubule-binding protein) homolog (PTRN-1) in *Caenorhabditis elegans*, have been reported to be involved in non-centrosomal microtubule formation (*Goodwin and Vale, 2010*; *Marcette et al., 2014*; *Meng et al., 2008*; *Richardson et al., 2014*; *Wu et al., 2016*). Although several in vitro reconstitution experiments have shown differences among CAMSAPs, CAMSAPs generally bind to the growing microtubule minus-ends and strongly suppress the dynamicity of non-centrosomal microtubules in different cell types, including neurons and epithelial cells (*Chuang et al., 2014*; *Hannak et al., 2002*; *Jiang et al., 2014*; *Martin et al., 2018*; *Nashchekin et al., 2016*; *Noordstra et al., 2016*; *Pongrakhananon et al., 2018*; *Sampaio et al., 2001*; *Tanaka et al., 2012*; *Toya et al., 2016*; *Wu et al., 2016*; *Yau et al., 2014*). The minus-end side of the microtubule lattice, but not the very end, is recognized by the CKK domain, although some additional effects were detected by other regions of CAMSAPs (*Atherton et al., 2019*; *Atherton et al., 2017*). CAMSAPs have been reported to act in parallel with γ-TuRC-dependent microtubule nucleation, and the microtubule-severing protein katanin regulates the functions of the CAMSAPs (*Jiang et al., 2018*; *Wang et al., 2015*; *Wu et al., 2016*). CAMSAPs are rapidly recruited to and to decorate pre-existing nascent microtubule minus-ends to sustain non-centrosomal microtubules. On the other hand, some reports have demonstrated

the possibility that CAMSAP-containing foci are involved in promoting microtubule nucleation independently of γ-TuRC (*Atherton et al., 2019*; *Jiang et al., 2018*; *Nashchekin et al., 2016*). Thus, the mechanism by which CAMSAP2 proteins contribute to non-centrosomal microtubule organization has remained controversial.

Recent studies suggest that microtubules in cells often nucleate independently of the γ-TuRC template (*Hannak et al., 2002*; *O'Toole et al., 2012*; *Sampaio et al., 2001*; *Tsuchiya and Goshima, 2021*; *Yau et al., 2014*). Therefore, we examined the roles of CAMSAPs in microtubule nucleation and microtubule network formation in vitro. In this study, we discovered the fundamental role of CAMSAP2 as a strong nucleator of microtubule formation from soluble αβ-tubulins. Our studies demonstrated that CAMSAP2 co-condenses with tubulins to stimulate spontaneous nucleation of microtubules without the γ-TuRC template. Many microtubule nucleation intermediates were observed inside the CAMSAP2-tubulin condensate during the early stages of microtubule polymerization. The CAMSAP2-tubulin condensate grows into the nucleation centre and mediates aster-like microtubule network formation in vitro, demonstrating the importance of CAMSAP2 in non-centrosomal, γ-tubulin-independent microtubule nucleation processes.

## Results

### Profiles of CAMSAP2 constructs used in in vitro assays

Initially, we examined the biochemical properties of the recombinant CAMSAP2 constructs used in this study by SDS-PAGE and size exclusion chromatography (*Figure 1A*). As shown in *Figure 1B and C*, full-length CAMSAP2 constructs with (GFP-CAMSAP2-FL) and without GFP-tag (CAMSAP2-FL) were purified to show a single peak in size exclusion chromatography. GFP-CAMSAP2-FL was eluted slightly earlier than the non-tagged construct due to the size of the GFP-tag, indicating that both constructs should be in the same oligomeric state. The molecular mass of CAMSAP2-FL was further examined by multi-angle light scattering (MALS) as 122 ± 1.1 kDa (*Figure 1—figure supplement 1*). Considering that the estimated molecular weight of CAMSAP2-FL is 167 kDa, it likely exists as a monomer in the solution.

We then analysed the microtubule binding pattern of GFP-CAMSAP2-FL by total internal reflection fluorescence (TIRF) microscopy. The results showed that GFP-CAMSAP2-FL bound to the microtubule lattice preferentially via the minus-ends, consistent with previous reports (*Atherton et al., 2017*; *Jiang et al., 2014*; *Figure 1D*). These observations collectively indicate that the CAMSAP2 proteins used in this study retained their original ability to bind to the microtubule minus-ends.

### CAMSAP2 reduces the microtubule nucleation barrier

Microtubule nucleation from αβ-tubulin heterodimers (hereafter called tubulin in the Results section) can occur spontaneously in vitro (*Kuchnir and Leibler, 1995*; *Voter and Erickson, 1984*). De novo microtubule nucleation in vitro requires a critical tubulin concentration greater than 20 μM ($Cc_{MT\ nucleation}$) (*Wieczorek et al., 2015*). This concentration is considerably higher than that required for microtubule elongation from seeds, which is typically approximately 1–2 μM and is called the critical concentration for microtubule polymerization ($Cc_{MT\ polymerization}$).

To test the nucleation ability of the purified pig tubulins used in this study, we performed a spin-down spontaneous nucleation assay through microtubule co-pelleting experiments in vitro (*Wieczorek et al., 2015*). We slightly modified the protocol to detect the functional CAMSAP2-FL; it can bind to microtubules at 37°C and become re-solubilized on ice along with the microtubule depolymerization. Nucleation was induced by 60 min of incubation at 37°C, accompanied by high-speed centrifugation. The pellets with or without CAMSAP2-FL were re-suspended and incubated on ice for 30 min for depolymerization. The samples were centrifuged again to eliminate non-specific aggregation, and the supernatant was applied for SDS-PAGE analysis (*Figure 2A*). We detected de novo nucleation of microtubules ($Cc_{MT\ nucleation}$) just above the tubulin concentration of 20 μM, which is consistent with a previous report and our result by pelleting assay without modification (*Wieczorek et al., 2015*; *Figure 2A and C*, and *Figure 2—figure supplement 1*).

We then evaluated the effect of CAMSAP2 on microtubule polymerization by a co-pelleting assay. We found that tubulin can be pelleted from 1.0 to 2.0 μM in the presence of 1.0 μM CAMSAP2-FL, indicating that tubulins could polymerize into microtubules or tubulin oligomers at a close concentration

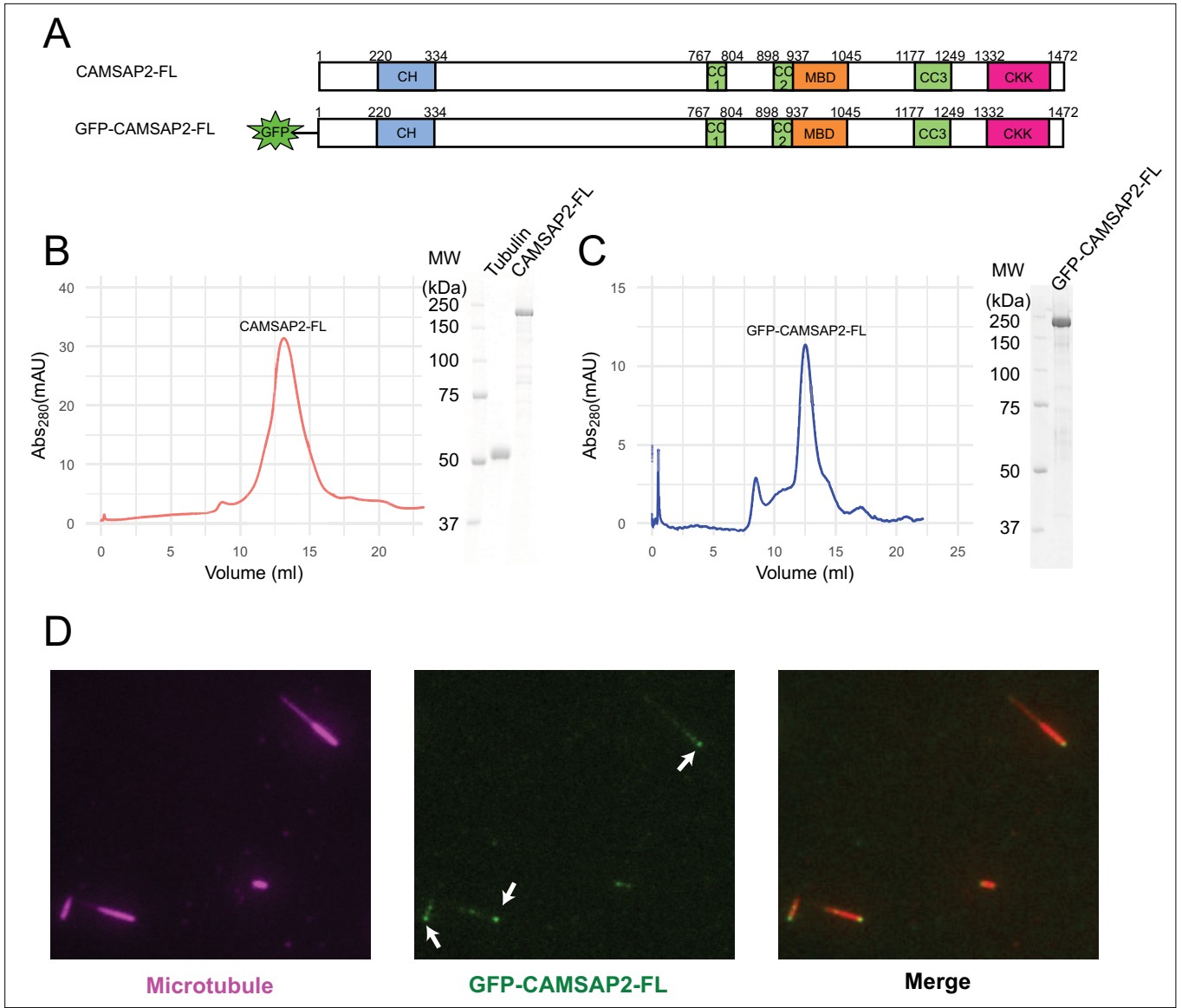

**Figure 1.** Functional study of recombinant calmodulin-regulated spectrin-associated protein 2 (CAMSAP2). (**A**) Schematic representation of the full-length CAMSAP2 constructs used in this study. CH, calponin-homology domain; MBD, microtubule-binding domain; CC, coiled-coil domain; CKK, C-terminal domain common to CAMSAP1 and two other mammalian proteins, KIAA1078 and KIAA1543. (**B**) (**C**) Size exclusion chromatography and SDS-PAGE of the peak fraction of (**B**) full-length CAMSAP2 and (**C**) GFP-CAMSAP2. (**D**) Total internal reflection fluorescence images of polarity-marked microtubules (magenta) decorated with purified GFP-CAMSAP2 (green). The minus-end segment of the microtubule is brighter than the plus-end segment.

The online version of this article includes the following source data and figure supplement(s) for figure 1:

**Source data 1.** *Figure 1* SDS-PAGE gel of the full-length calmodulin-regulated spectrin-associated protein 2.

**Figure supplement 1.** Size exclusion chromatography with multi-angle light scattering of the calmodulin-regulated spectrin-associated protein 2-FL.

to the $Cc_{MT\ polymerization}$ with CAMSAP2-FL (*Figure 2B and C*). CAMSAP2 drastically reduced the nucleation barrier to a level close to the theoretical limit of microtubule polymerization. In this assay, both tubulin and CAMSAP2-FL in the pellet were recovered from the supernatant after incubation on ice, suggesting that CAMSAP2-FL co-precipitated with tubulins and this precipitation is reversible.

We further tested microtubule polymerization at the different concentrations of the CAMSAP2-FL from 10 nM to 1.0 μM in the tubulin polymerization buffer (PEM) or in the PEM supplemented with 100 mM KCl, which mimics a physiological condition. Polymerization was detected from 250 nM CAMSAP2-FL in PEM, and 500 nM CAMSAP2-FL in PEM +100 mM KCl, albeit the polymerization

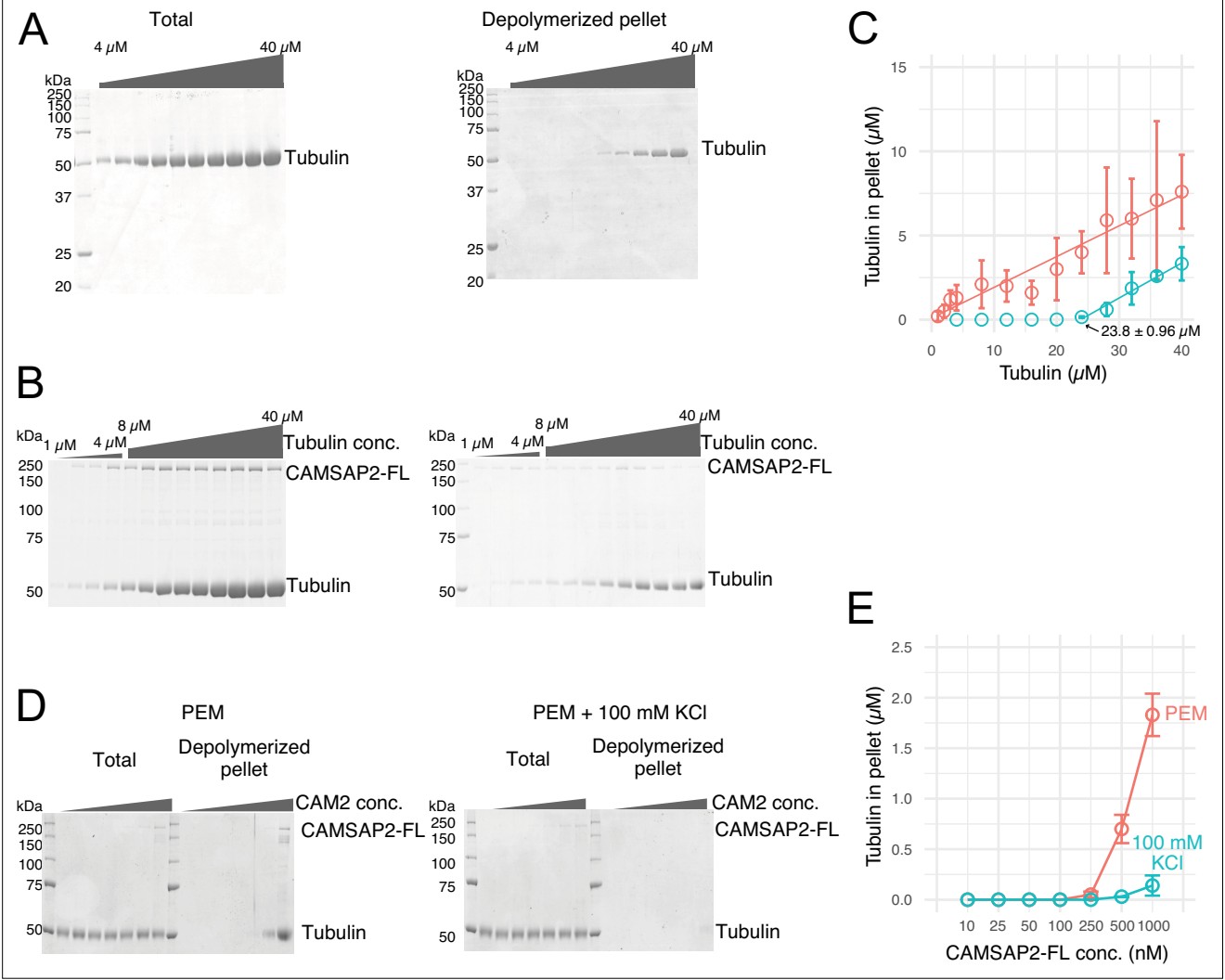

**Figure 2.** Calmodulin-regulated spectrin-associated protein 2 (CAMSAP2) stimulates microtubule nucleation. (**A**) SDS-PAGE gels from a spin-down spontaneous nucleation assay of tubulin showing the total tubulin after 60 min of polymerization at 37°C (left gel). Polymerized tubulin was pelleted by centrifugation and then depolymerized on ice and centrifuged to remove debris, and the supernatant was subjected to SDS-PAGE (right gel). (**B**) SDS-PAGE gels from a spin-down spontaneous nucleation assay of tubulin with 1 μM CAMSAP2-FL showing the total tubulin-CAMSAP2 (left gel) and the polymerized/depolymerized tubulin (right gel). (**C**) Plots of the depolymerized tubulin concentrations determined by pelleting assay against the total tubulin concentrations determined by the reaction on SDS-PAGE gel from three independent assays (mean ± SD). The depolymerized tubulin from the tubulin assay is turquoise green, and that combined with CAMSAP2-FL is orange. The concentrations of tubulin greater than 0.1 μM are fitted with a trend line that has an x-intercept of 23.8 ± 0.96 μM (Cc_MT nucleation). (**D**) SDS-PAGE gels from a spin-down spontaneous nucleation assay of 10 μM tubulin with 10–1000 nM CAMSAP2-FL in the PEM (100 mM PIPES pH 6.8, 1 mM MgCl$_2$, 1 mM EGTA, and 1 mM GTP) or PEM + 100 mM KCl. (**E**) Plots of the depolymerized pelleting assay in the different concentration of the CAMSAP2-FL with different buffer with SD from three independent assays (mean ± SD).

The online version of this article includes the following source data and figure supplement(s) for figure 2:

**Source data 1.** SDS-PAGE gel from a spin-down spontaneous nucleation assay of tubulin showing the total tubulin after 60 min of polymerization at 37°C.

**Source data 2.** SDS-PAGE gel of depolymerized tubulin after spin-down pelleting assay.

**Source data 3.** SDS-PAGE gel from a spin-down spontaneous nucleation assay of tubulin with 1 μM calmodulin-regulated spectrin-associated protein 2-FL showing the total tubulin after 60 min of polymerization at 37°C.

**Source data 4.** SDS-PAGE gel of depolymerized tubulin with 1 μM calmodulin-regulated spectrin-associated protein 2-FL after spin-down pelleting assay.

**Source data 5.** Quantification of the depolymerized tubulin concentrations determined by pelleting assay.

**Source data 6.** SDS-PAGE gel from a spin-down spontaneous nucleation assay of 10 μM tubulin with 10–1000 nM calmodulin-regulated spectrin-

*Figure 2 continued on next page*

*Figure 2 continued*

associated protein 2-FL in the PEM.

**Source data 7.** SDS-PAGE gel from a spin-down spontaneous nucleation assay of 10 µM tubulin with 10–1000 nM calmodulin-regulated spectrin-associated protein 2-FL in the PEM + 100 mM KCl.

**Source data 8.** Quantification of the depolymerized tubulin concentrations determined by pelleting assay.

**Figure supplement 1.** Comparison of pelleting assay.

**Figure supplement 1—source data 1.** SDS-PAGE gel of tubulin spin-down pelleting assay without depolymerization (no depoly).

**Figure supplement 1—source data 2.** Quantification of the depolymerized tubulin concentrations determined by pelleting assay.

efficiency considerably decreased by increasing the salt concentration (*Figure 2D and E*). These results suggest that CAMSAP2 significantly enhances the nucleation and polymerization of tubulin. The enhancement effect still exists in the physiological condition, although it became milder than the tubulin polymerization buffer.

## CAMSAP2 recruits tubulins through liquid-liquid phase separation

Recent studies of microtubule-associated proteins, TPX2 and Tau, showed that these proteins undergo liquid-liquid phase separation (LLPS) to interact with microtubules through an internal long disordered region (*King and Petry, 2020*; *Tan et al., 2019*). CAMSAP2 also possesses a long disordered region at the middle portion, as predicted by program PONDR, indicating its potential for phase separation (*Figure 3A*; *Romero et al., 2001*). We thus investigated the droplet formation of CAMSAP2 in the polymerization PEM buffer, finding that both CAMSAP2-FL and GFP-CAMSAP2-FL formed droplets (*Figure 3B*). We next evaluated the concentration of CAMSAP2 or salt on the droplet formation. By examining a series of different GFP-CAMSAP2-FL and salt concentrations, we found that 10 nM GFP-CAMSAP2-FL, which approximates a physiological concentration, could form the droplet under the salt concentration of less than 400 mM, which is similar to TPX2 (*King and Petry, 2020*; *Figure 3C and D*).

To evaluate the nature of the CAMSAP2 condensate, we observed their liquid fluidity. We increased the concentration of GFP-CAMSAP2-FL to 4 µM to observe condensates efficiently. Once two CAMSAP2 condensates came close to each other, they fused and became a single large condensate (*Figure 3E*, *Video 1*). We also observed the fluorescence recovery after photobleaching (FRAP) in a CAMSAP2 condensate and found a slow recovery of the fluorescence (*Figure 3F and G*, *Figure 3—figure supplement 1*, and *Video 2*). These results suggest that the CAMSAP2 condensates possess high viscosity.

We then tested whether CAMSAP2 can recruit tubulins into its condensate. In the physiological condition (PEM with 100 mM KCl), tubulin alone did not form any droplet but formed co-condensate only when mixed with GFP-CAMSAP2-FL (*Figure 3H*). These findings suggest that CAMSAP2 forms condensates by phase separation mechanism, into which tubulins are recruited.

## CAMSAP2 and tubulin co-condense to form aster-like structure

To investigate how CAMSAP2 regulates the nucleation and polymerization of microtubules in connection with phase separation, we co-condensed tubulin and GFP-CAMSAP2-FL and observed them by fluorescence microscopy. Ten micromolar tetramethylrhodamine (TMR)-labelled tubulin was co-incubated with 10–1000 nM full-length GFP-CAMSAP2-FL at 37°C for 10 min in the physiological condition (PEM with 100 mM KCl) (*Figure 4A*). These molecules co-condensed together, and importantly microtubules extended radially from the condensates (*Figure 4B*), as observed for the TPX2 aster-like structure (*Roostalu et al., 2015*; *Schatz et al., 2003*). The appearance of this structure was reminiscent of the centrosomal aster; thus, we named it 'Cam2-aster'. Cam2-asters became detectable above 250 nM of GFP-CAMSAP2-FL, and the number of asters increased as the CAMSAP2 concentration raised from 250 to 1000 nM, consistent with our pelleting assay (*Figure 3B and D*).

The CAMSAP2 concentration required for aster formation in vitro was higher than the putative physiological concentration of CAMSAP2 (tens of nanomolar) (*Wühr et al., 2014*). Previous reports have demonstrated that CAMSAPs are not only cytosolic but also concentrated at sub-cellular structures such as adherens junctions and cellular cortex through the interaction with some adapter proteins (*Toya et al., 2016*), and these CAMSAPs that are anchored to semi-solid structures sustained

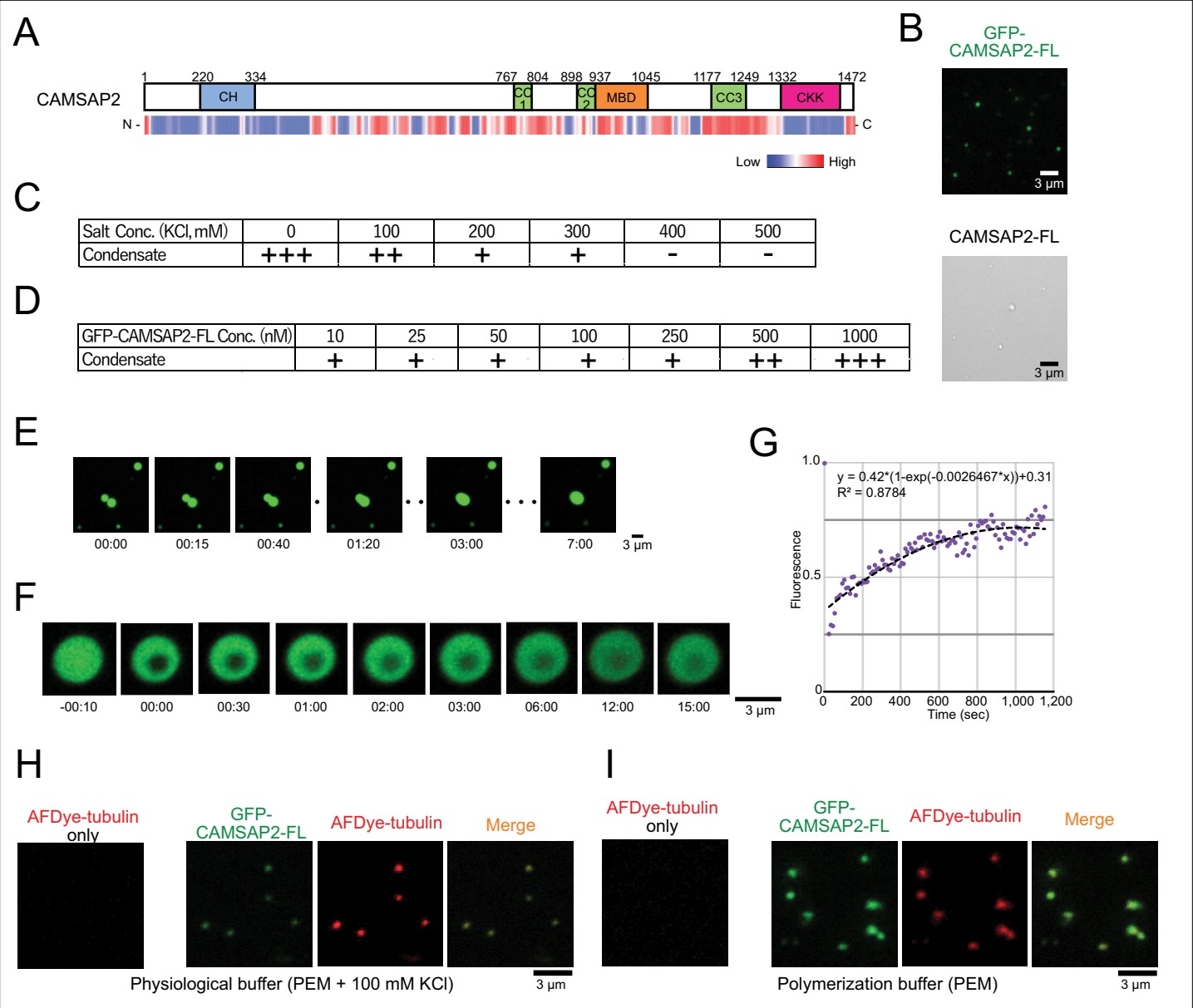

**Figure 3.** Calmodulin-regulated spectrin-associated protein 2 (CAMSAP2) forms co-condensate with tubulin in vitro. (**A**) Intrinsic disorder prediction of CAMSAP2 by PONDR. CH, calponin-homology domain; MBD, microtubule-binding domain; CC, coiled-coil domain; CKK, C-terminal domain common to CAMSAP1 and two other mammalian proteins, KIAA1078 and KIAA1543. (**B**) Fluorescent image of GFP-CAMSAP2-FL condensates (top) and DIC image of CAMSAP2-FL condensates (bottom). (**C, D**) Phase diagram of 1 µM GFP-CAMSAP2-FL with indicated salt concentrations (**C**) and different concentrations of GFP-CAMSAP2-FL with 100 mM KCl (**D**). (**E**) Fusion of the GFP-CAMSAP2-FL condensate (also see *Video 1*). (**F**) Fluorescence recovery after photobleaching of GFP-CAMSAP2-FL condensates, acquired via confocal microscopy and (**G**) quantification. Time 00:00 (minutes:seconds) corresponds immediately after photobleaching. The graph shows the fluorescence recovery process of one of the four quantified droplets in *Figure 3—figure supplement 1*. (**H**) (**I**) GFP-CAMSAP2-FL and tubulin formed co-condensate in the physiological buffer (PEM with 100 mM KCl) and microtubule polymerization buffer (PEM).

The online version of this article includes the following source data and figure supplement(s) for figure 3:

**Source data 1.** Quantification of the fluorescence recovery after photobleaching of GFP-calmodulin-regulated spectrin-associated protein 2-FL condensates, acquired via confocal microscopy.

**Figure supplement 1.** Three different measurements of fluorescence recovery after photobleaching of GFP-calmodulin-regulated spectrin-associated protein 2-FL condensates, acquired via confocal microscopy and quantification.

**Figure supplement 1—source data 1.** Quantification of the fluorescence recovery after photobleaching of GFP-calmodulin-regulated spectrin-associated protein 2-FL condensates, acquired via confocal microscopy.

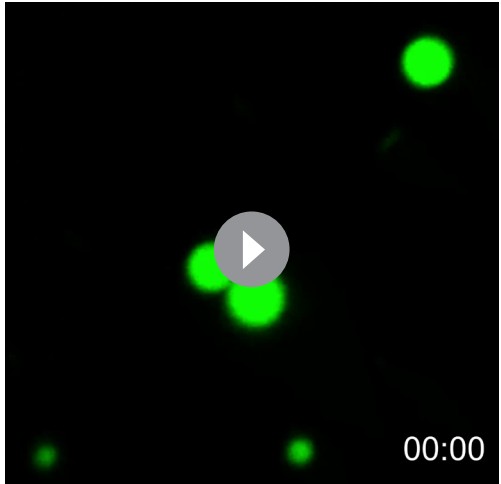

**Video 1.** Fusion of the calmodulin-regulated spectrin-associated protein 2 (CAMSAP2) condensate.
https://elifesciences.org/articles/77365/figures#video1

microtubule growth. We also observed the local condensates of intrinsic CAMSAP2 in HeLa cells before and during the polymerization of microtubules (*Figure 4—figure supplement 1*). We thus thought that the physical stabilization of CAMSAP2 might increase the efficiency of microtubule polymerization. To test this hypothesis, we physically fixed GFP-CAMSAP2-FL on the coverslip surface with the condition in which GFP-CAMSAP2-FL forms condensates (*Figure 3*). Biotinylated GFP antibody was fixed on the coverslip through the streptavidin binding. Preformed GFP-CAMSAP2 condensates were then applied (*Figure 4D*), followed by the addition of TMR-labelled tubulin, and visualized by TIRF microscopy. Tubulin was efficiently incorporated into the CAMSAP2 foci within 1 min, although polymerization of microtubules was rarely observed (*Figure 4E*). We thought that microtubule polymerization from CAMSAP2 foci might require free CAMSAP2 in solution, and we supplemented

10–1000 nM of free GFP-CAMSAP2-FL (*Figure 4F*). As a consequence, microtubule formation was induced from the CAMSAP2 foci. GFP-CAMSAP2-FL at a concentration of 50 nM in solution was sufficient for efficient microtubule polymerization from the foci, although higher concentrations of GFP-CAMSAP2-FL further increased the microtubule polymerization efficiency (*Figure 4G and H*). Thus, when CAMSAP2 is physically fixed as observed in vivo, its concentration required for microtubule elongation could be reduced.

Next, we acquired time-lapse images of microtubules using GFP-CAMSAP2-FL and TMR-labelled tubulin by TIRF microscopy. We found that dynamic polymerization and depolymerization of microtubules occurred at CAMSAP2-containing foci with the association of GFP-CAMSAP2-FL (*Figure 4I*, *Videos 3 and 4*). These findings suggest that the fixed foci of CAMSAP2 act as a core for microtubule polymerization, with free CAMSAP2 leading to the formation of microtubule asters.

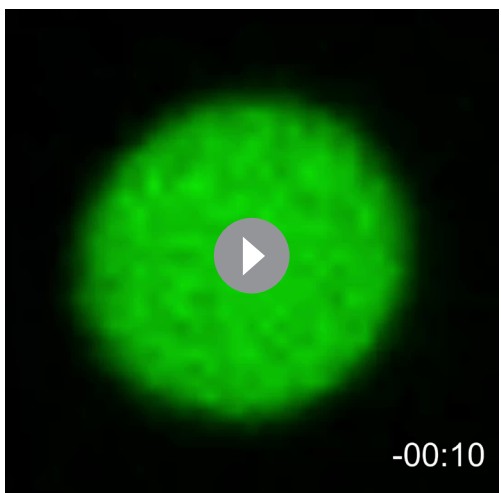

**Video 2.** Fluorescence recovery after photobleaching of GFP-calmodulin-regulated spectrin-associated protein 2-FL condensates, acquired via confocal microscopy. The video was recorded immediately after photobleaching.
https://elifesciences.org/articles/77365/figures#video2

## Visualization of Cam2-asters by electron microscopy

To visualize structural details of Cam2-aster formation, we employed electron microscopic analysis. We first observed condensates of CAMSAP2-FL by negative stain electron microscopy (EM) (*Figure 5A*). Consistent with light microscopy observations, the spherical CAMSAP2 condensates were detected. Next, we mixed tubulin and CAMSAP2-FL and observed them by negative stain EM in various conditions. When 10 μM tubulin was incubated with 1 μM CAMSAP2-FL at 37°C for 30 min, microtubule bundles radiated extensively from each condensate (*Figure 5B*), as found in the TIRF experiments (*Figure 4H*). Microtubule bundles from the neighbouring condensates further interacted, forming a dense microtubule meshwork (*Figure 5B*). CAMSAP2 homolog CAMSAP3, which is reported to stimulate microtubule nucleation and growth in vitro (*Roostalu et al., 2018*), also exhibits the ability to promote the microtubule aster (*Figure 5—figure*

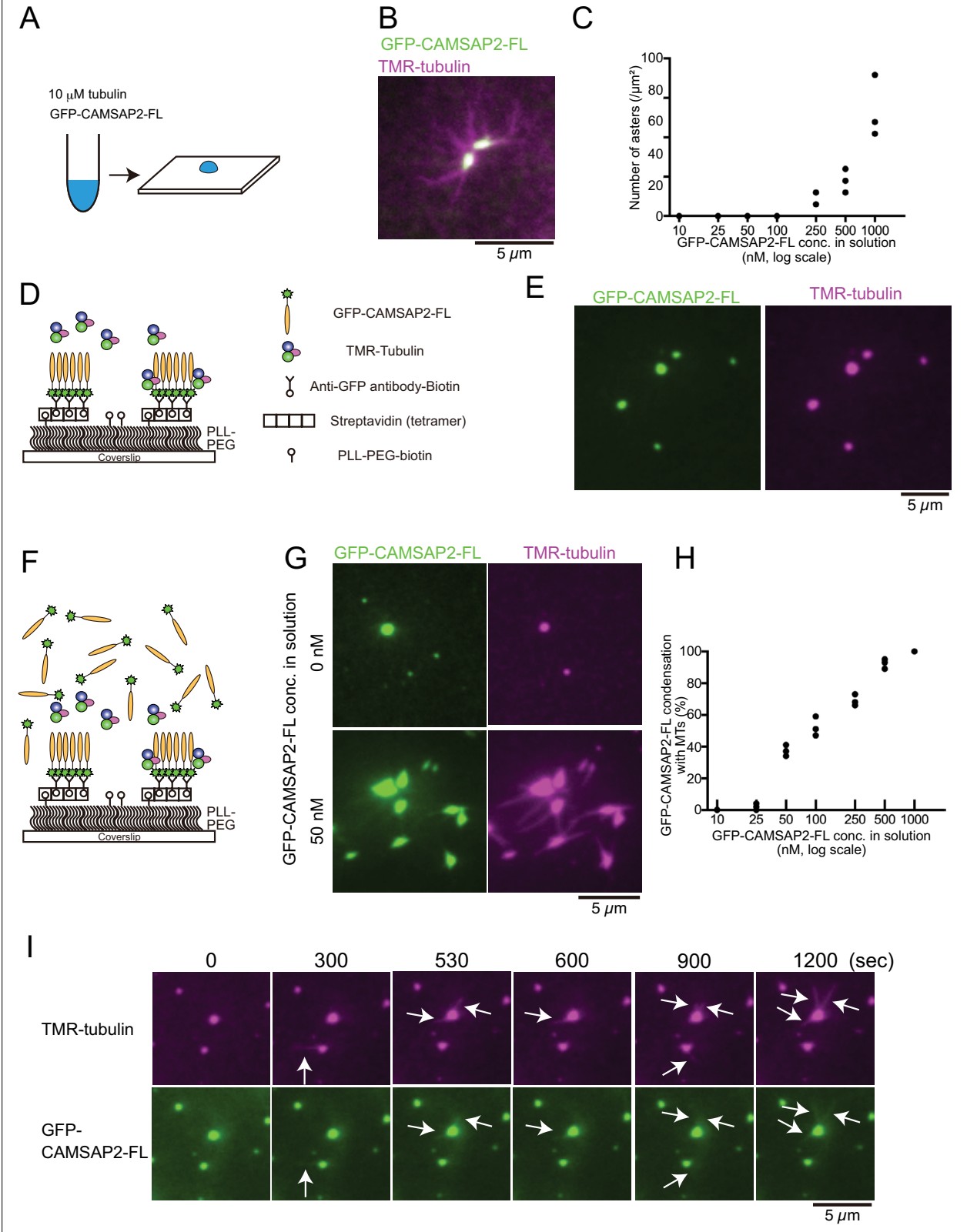

**Figure 4.** Tubulin is incorporated into calmodulin-regulated spectrin-associated protein 2 (CAMSAP2) condensates to form aster-like structure. (**A**) The procedure used to obtain the data in panels (**B**) and (**C**). GFP-CAMSAP2-FL, 10 μM tubulin, and 0.5 μM tetramethylrhodamine (TMR)-tubulin were mixed in BRB80 supplemented with 100 mM KCl and incubated for 10 min at 37°C. The solution was directly transferred onto a coverslip and observed by fluorescence microscopy. (**B**) Representative image of asters. GFP-CAMSAP2-FL (0.5 μM), tubulin (10 μM), and TMR-tubulin (0.5 μM) were co-incubated.

*Figure 4 continued on next page*

*Figure 4 continued*

(**C**) Quantification of the numbers of asters in solutions containing 10 µM tubulin, 0.5 µM TMR-tubulin, and 10, 25, 50, 100, 250, 500, and 1000 nM GFP-CAMSAP2-FL. The results of three independent assays are shown with dots. (**D**) Schematic showing reconstitution of CAMSAP2-containing foci. CAMSAP2 condensates were formed as described in *Figure 3* and fixed on the coverslip by an anti-GFP antibody. (**E**) Soluble tubulins were incorporated into CAMSAP2 condensates within 1 min. (**F**) Schematic showing CAMSAP2-containing foci with soluble tubulin and GFP-CAMSAP2-FL. (**G**) CAMSAP2 in solution induced aster formation from CAMSAP2 condensates in a dose-dependent manner. Representative images for 0 and 50 nM GFP-CAMSAP2-FL are shown. (**H**) Quantification of microtubule formation from CAMSAP2 condensates. The percentages of CAMSAP2 condensates with microtubules among total CAMSAP2 condensates are shown. Each dot shows the results of three independent experiments. (**I**) Time-lapse images of aster formation. Tubulin (10 µM), TMR-tubulin (0.5 µM), and GFP-CAMSAP2-FL (50 nM) were incubated with CAMSAP2 condensates fixed on coverslips. Dynamic microtubules from CAMSAP2 condensates were observed (arrows). The scale bars indicate 5 µm. See *Videos 3 and 4*.

The online version of this article includes the following source data and figure supplement(s) for figure 4:

**Figure supplement 1.** Calmodulin-regulated spectrin-associated protein 2 (CAMSAP2) localization in growing microtubule networks in HeLa cells.

**Figure supplement 1—source data 1.** Original image of immunofluorescence visualized by a confocal laser-scanning microscope.

*supplement 1*), representing the aster formation activity might be a common function among CAMSAPs.

To test the tubulin concentration dependency of Cam2-aster formation, we used 2 µM tubulin in mixing with 1 µM CAMSAP2-FL at 37°C for 30 min, which is the minimum tubulin concentration that can induce pellet formation (*Figure 2B*). Consequently, we found a similar microtubule meshwork although in fewer numbers (*Figure 5—figure supplement 2A*). We also observed 10 µM tubulin with different concentrations of CAMSAP2-FL from 250 nM to 1 µM in the PEM or the physiological buffer (100 mM KCl + PEM) at 37°C (*Figure 5—figure supplement 2B, C*). All except for the case with 250 nM CAMSAP2-FL in the physiological buffer produced aster and microtubule meshwork. We should also note that the increase in salt concentrations decreased the number of microtubules extending from the condensate (*Figure 5—figure supplement 2B, C*).

To eliminate the possible artifacts induced by negative staining, we examined the samples by cryo-EM. To efficiently visualize the aster in cryo condition, we increased the concentrations of tubulin and CAMSAP2-FL to 30 µM and 3 µM, respectively, and co-incubated them at 37°C for 10 min. Since 30 µM of tubulin exceeded the $Cc_{MT\ nucleation}$ (*Figure 2C*), we first confirmed that, in this experimental condition, microtubules were rarely observed after incubation of the tubulins without CAMSAP2 on ice, and at 37°C for 1 min, 3 min, and 10 min (*Figure 5—figure supplement 3*). When 30 µM tubulin was incubated with 3 µM CAMSAP2-FL, Cam-2 asters formed. Microtubule bundles were radially connected with CAMSAP2 condensates (*Figure 5C*), as observed in the TIRF and negative stain EM samples.

## Functional domain mapping for CAMSAP2 microtubule-nucleation and Cam2-aster formation

To identify the functional domain of CAMSAP2 involved in microtubule nucleation and Cam2-aster formation, we prepared a series of deletion constructs and tested their functions (*Figure 6A* and *Figure 6—figure supplements 1 and 2*). We first acquired the time-lapse images of microtubule formation using the GFP-labelled CAMSAP2

**Video 3.** Time-lapse movie of aster formation. Tubulin (10 µM), TMR-tubulin (0.5 µM), and GFP-calmodulin-regulated spectrin-associated protein 2 (CAMSAP2)-FL (50 nM) were incubated with CAMSAP2 condensates fixed on coverslips. The size of the field is 81.9 µm × 81.9 µm. The frame rate is 0.2 frames/s.

https://elifesciences.org/articles/77365/figures#video3

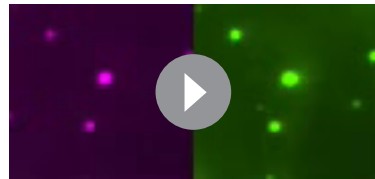

**Video 4.** Magnified movie of aster formation from 1. Dynamic microtubules from calmodulin-regulated spectrin-associated protein 2 condensates were observed. The size of the field is 8.2 µm × 8.2 µm. The frame rate is 0.2 frames/s.

https://elifesciences.org/articles/77365/figures#video4

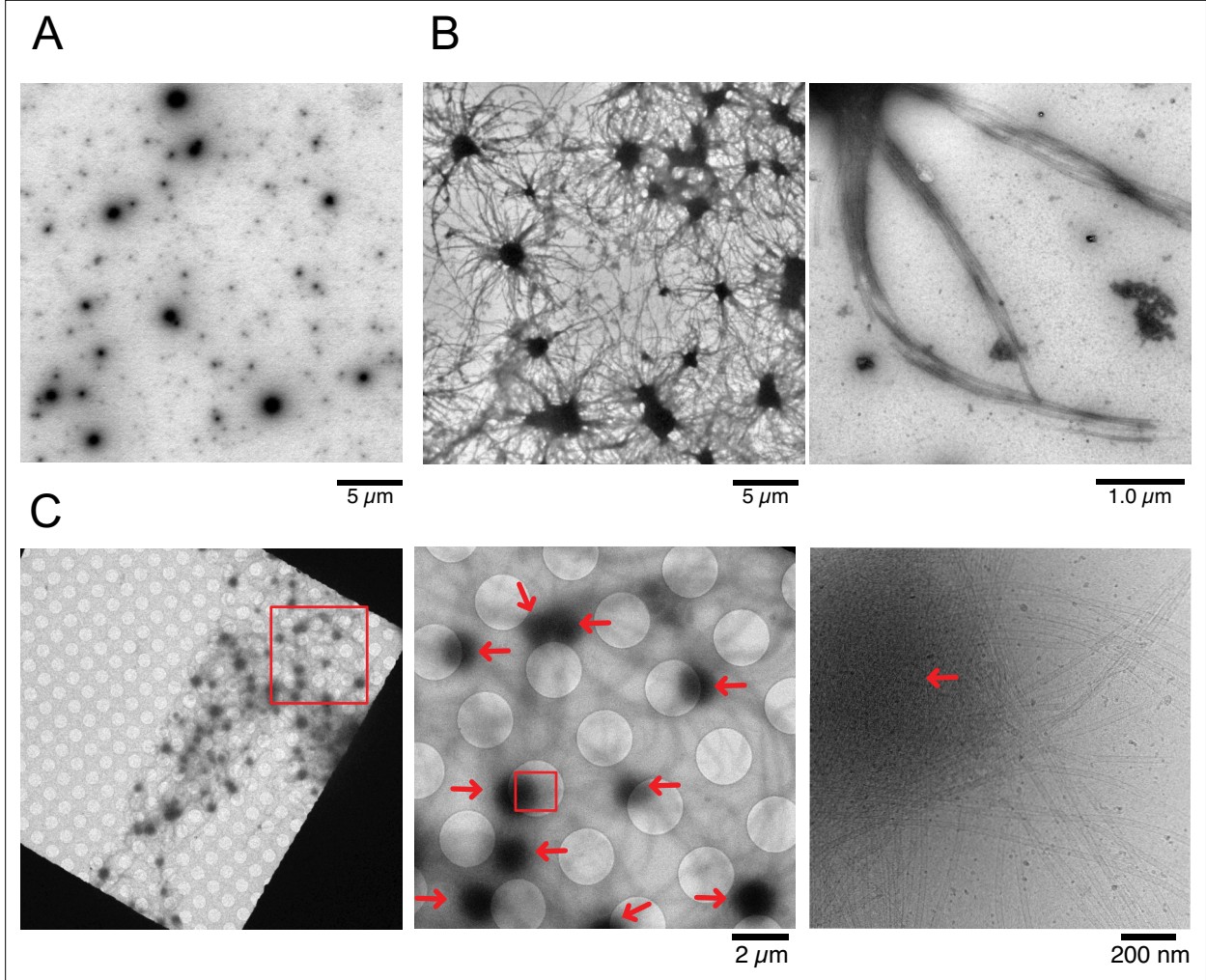

**Figure 5.** Nucleation and aster formation activity of calmodulin-regulated spectrin-associated protein 2 (CAMSAP2). Representative electron microscopy (EM) images are shown from at least three independent assays. (**A**) Negative stain EM micrographs of 1 μM CAMSAP2-FL incubated at 25°C for 30 min. (**B**) Negative stain EM micrographs of 10 μM tubulin polymerized with 1 μM of CAMSAP2-FL after incubation at 37°C for 10 min. Aster-like microtubule structures were observed. Negative stain micrographs of tubulin with CAMSAP2-FL incubated at various conditions were also available in *Figure 5—figure supplement 2*. (**C**) Cryo-EM micrographs of 30 μM tubulin polymerized with 3 μM CAMSAP2-FL after incubation at 37°C for 10 min captured at different magnifications. Cam2-asters are indicated by the red arrows. The cryo-EM micrographs of 30 μM tubulin on ice, polymerized for 1, 3, and 10 min at 37°C are available in *Figure 5—figure supplement 3*.

The online version of this article includes the following source data and figure supplement(s) for figure 5:

**Figure supplement 1.** Aster formation activity of recombinant calmodulin-regulated spectrin-associated protein 3 (CAMSAP3).

**Figure supplement 1—source data 1.** SDS-PAGE gels of full-length GFP-CAMSAP3.

**Figure supplement 2.** Nucleation and aster formation activity of calmodulin-regulated spectrin-associated protein 2 (CAMSAP2).

**Figure supplement 3.** The cryo-electron microscopy micrographs of 30 μM tubulin at different conditions.

deletion constructs and TMR-labelled tubulin by TIRF microscopy (*Figure 6B*), utilizing the same strategy in *Figure 4F*. Consequently, GFP-CC3-CKK and GFP-CC1-CKK could induce the Cam2-aster formation, whereas no condensate or aster was found with GFP-CKK (*Figure 6B and C*). The fluorescence intensities of condensates at the centre of Cam2-asters were the highest in GFP-CAMSAP2-FL and the lowest in GFP-CC3-CKK (*Figure 6B and C*). Therefore, the N-terminal region preceding the CC3 is vital for the condensate formation, consistent with the intrinsic disorder region presented in *Figure 3A*. We should note that GFP-CC1-CKK and GFP-CC3-CKK, whose 18 amino acid residues at the C-terminus are deleted, were utilized for the experiments because they were functional and more stable than GFP-CC1-CKK-CT or GFP-CC3-CKK-CT (*Figure 6—figure supplement 2*).

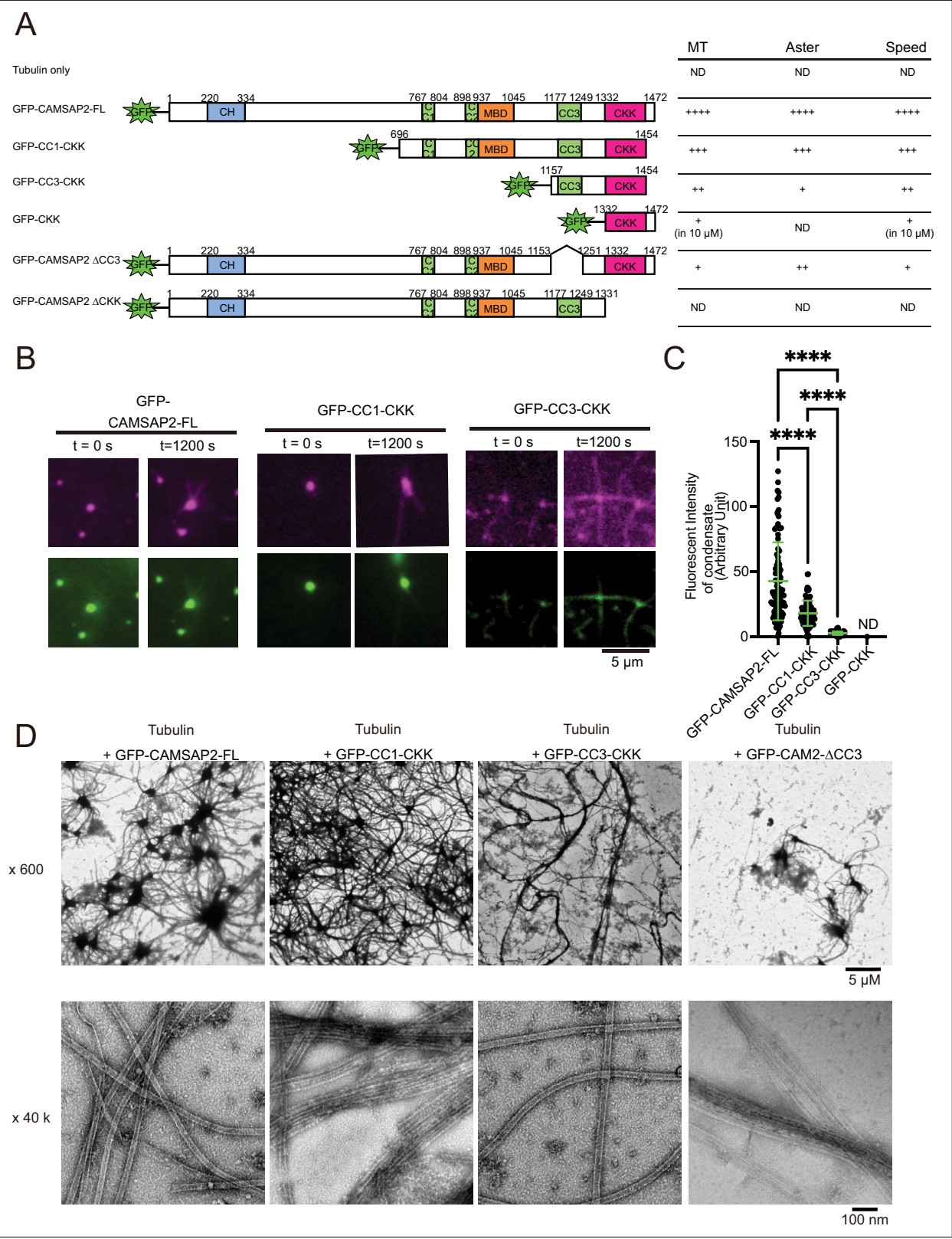

**Figure 6.** Functional domain mapping of the microtubule nucleation and aster formation activity of calmodulin-regulated spectrin-associated protein 2 (CAMSAP2). (**A**) Microtubule nucleation and aster formation activities of CAMSAP2 deletion constructs evaluated by the results of 10 μM tubulin with 1 μM CAMSAP2. The number of '+' symbols indicates the strength of the activity (++++, strongest; +, weakest; ND, not detected). Size exclusion chromatography and SDS-PAGE of GFP fused constructs are available in *Figure 6—figure supplement 1*. CH, calponin-homology domain; MBD,

*Figure 6 continued on next page*

*Figure 6 continued*

microtubule-binding domain; CC, coiled-coil domain; CKK, C-terminal domain common to CAMSAP1 and two other mammalian proteins, KIAA1078 and KIAA1543. (**B**) Microtubule growth from CAMSAP2 condensates composed of full-length and deletion constructs. In vitro reconstitution was performed as described in *Figure 4I*. (**C**) Fluorescent intensity of CAMSAP2 condensates at 0 s. ND means that the fluorescent intensity of condensates could not be measured because GFP-CKK did not induce any condensates. Ordinary one-way ANOVA followed by Tukey's multiple comparisons test. ****, p<0.0001. n=100 condensates from three independent preparations. (**D**) Negative stain EM images of polymerization by 10 μM tubulin with 1 μM GFP-CAMSAP2-FL or 1 μM CAMSAP2 mutants during 10 min of incubation at 37°C. The results for tubulin alone and GFP-CKK are available in *Figure 6—figure supplement 3*. Representative EM images are shown from at least three independent assays.

The online version of this article includes the following source data and figure supplement(s) for figure 6:

**Source data 1.** Quantifications of fluorescent intensity of calmodulin-regulated spectrin-associated protein 2 condensates at 0 s.

**Figure supplement 1.** GFP fused calmodulin-regulated spectrin-associated protein 2 (CAMSAP2) constructs used in this study.

**Figure supplement 1—source data 1.** SDS-PAGE gel of calmodulin-regulated spectrin-associated protein 2 deletion constructs.

**Figure supplement 1—source data 2.** SDS-PAGE gel of calmodulin-regulated spectrin-associated protein 2 deletion constructs.

**Figure supplement 2.** Functional domain mapping of the microtubule nucleation and aster formation activity of calmodulin-regulated spectrin-associated protein 2 (CAMSAP2) deletion mutants.

**Figure supplement 2—source data 1.** SDS-PAGE gel of calmodulin-regulated spectrin-associated protein 2 deletion constructs.

**Figure supplement 3.** Functional domain mapping of calmodulin-regulated spectrin-associated protein 2 (CAMSAP2) analysed by negative stain electron microscopy (EM).

We next used negative stain EM assays. 10 μM of tubulin below the $Cc_{MT\ nucleation}$ was mixed with each of the CAMSAP2 mutants (1 μM) on ice, incubated at 37°C for 10 min, and examined for microtubule nucleation and aster-forming activity (*Figure 6A and D*). We first confirmed that tubulin alone formed oligomers, but did not grow into microtubules after 10 min of incubation at 37°C (*Figure 6—figure supplement 3*), consistent with the results of the microtubule pelleting assay (*Figure 2*). Contrastingly, tubulin in the presence of GFP-CAMSAP2-FL produced a dense Cam2-aster network as described above (*Figure 6D*), which functions equivalent to CAMSAP2-FL. With GFP-CC1-CKK, the Cam2-aster network was similarly observed, although the condensates were smaller than those of GFP-CAMSAP2-FL (*Figure 6D*). GFP-CC3-CKK could also form Cam2-asters, but condensations were less represented and the microtubule meshwork was sparser (*Figure 6D*).

We then investigated the role of the CKK domain, which is the primary binding site for the microtubule. The CKK domain only (GFP-CKK) or a deletion construct of the CKK domain (GFP-CAMSAP2 ΔCKK) at 1 μM yielded few microtubules nor Cam2-asters but produced clumps of tubulin oligomers, consistent with the previous report (*Figure 6—figure supplement 3*; *Atherton et al., 2017*). GFP-CKK at 10 μM did not form any aster, albeit microtubules did form (*Figure 6—figure supplement 3B*). We also tested the ability for microtubule nucleation and formation of the CC3 domain by generating the GFP-CAMSAP2 ΔCC3. GFP-CAMSAP2 ΔCC3 could nucleate microtubules but a fewer number of asters (*Figure 6D*). The coiled-coil regions, at least the CC3 domain, thus seem to be required for microtubule polymerization and Cam2-aster formation. Collectively, CKK is essential and CC3 with CKK is the minimum requisites for microtubule nucleation and Cam2-aster formation by CAMSAP2.

## Ring structures observed during microtubule nucleation with CAMSAP2

Next, we aimed to analyse the process of generating the microtubule nucleation centre. We chose GFP-CC1-CKK for this analysis because the centre of Cam2-asters induced by full-length CAMSAP2-FL or GFP-CAMSAP2-FL was too thick for TEM observation. GFP-CC1-CKK showed a similar microtubule nucleation ability with a less condensed nucleation centre (*Figure 6D*). We examined the time-dependent process of microtubule nucleation and aster formation by negative stain EM. Incubation of 10 μM tubulin with 1 μM GFP-CC1-CKK for 30 min on ice induced the formation of numerous rings with diameters of approximately 50 nm (*Figure 7A*); such rings were never detected in the tubulin-alone on ice condition (*Figure 6—figure supplement 3A*). At 1 min after 37°C incubation, many short microtubules appeared with a concomitant reduction of the rings. Further incubation at 37°C increased the microtubule number and length, but decreased the number of rings (*Figure 7A and B*). The ring number was the highest when incubation was conducted on ice, and rings eventually disappeared after 10 min incubation at 37°C, suggesting the rings would be utilized for some processes involved in microtubule growth.

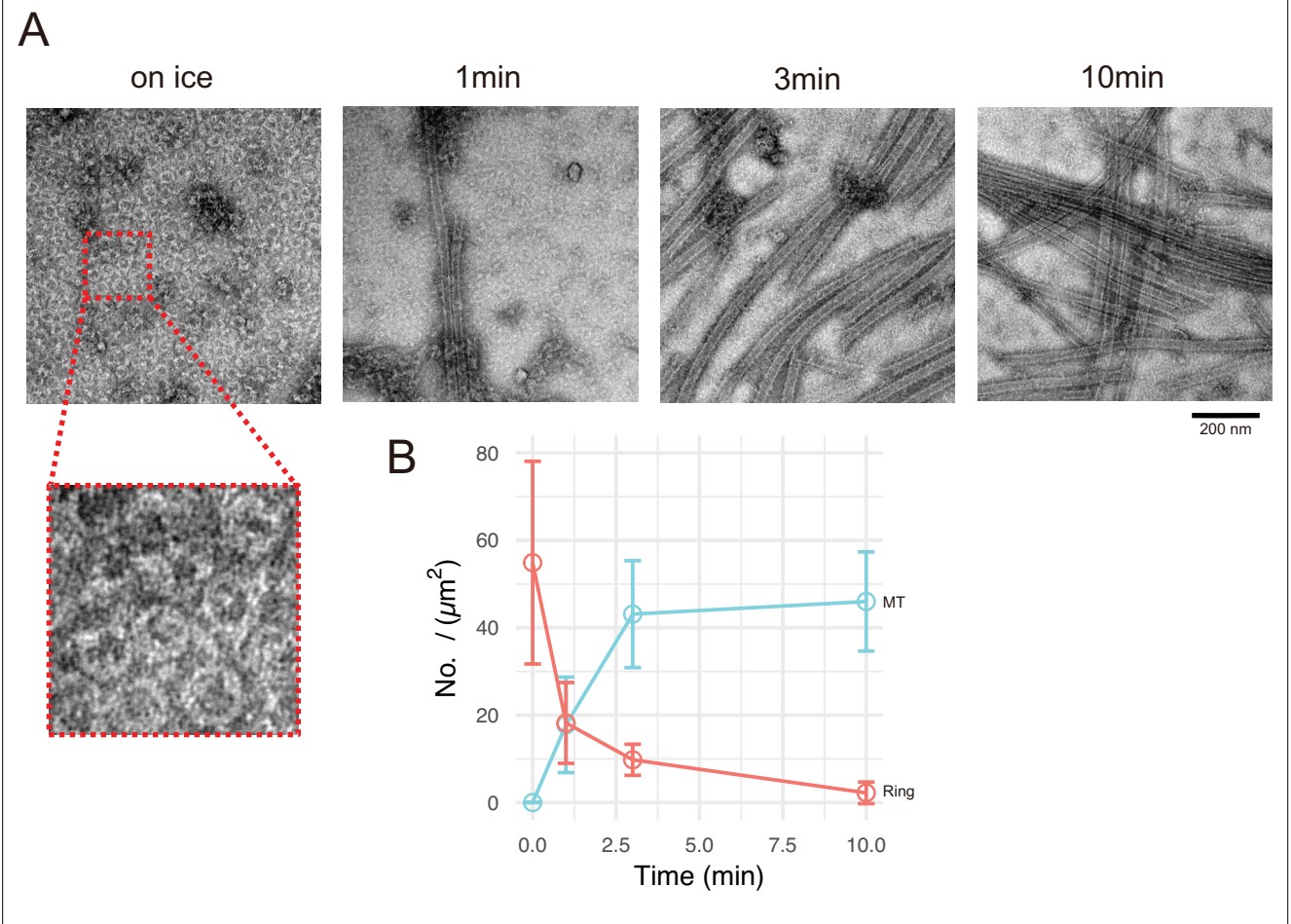

**Figure 7.** Calmodulin-regulated spectrin-associated protein 2 (CAMSAP2) induces tubulin ring formation. Representative electron microscopy (EM) images are shown from at least three independent assays. (**A**) Negative stain EM micrographs of 10 µM tubulin polymerization with 1 µM GFP-CC1-CKK at different time points. (**B**) Plots of the number of tubulin rings (orange) and that of microtubules (cyan) at different time points (mean ± SD, from 10 independent views).

The online version of this article includes the following source data for figure 7:

**Source data 1.** Quantification of the numbers of rings and microtubules at different time points.

## CAMSAP2-induced microtubule nucleation intermediates observed by cryo-EM

We further observed the process of Cam2-aster growth by cryo-EM at three different time points. We incubated the tubulin–GFP-CC1-CKK mixture on ice for 30 min, at 37°C for 1 min, and 3 min. Tubulin and GFP-CC1-CKK concentrations were set to 30 µM and 3 µM, respectively; the ratio was identical to that used for negative stain EM, but the concentration of each molecule was tripled to observe the rings or sheets more efficiently. Because 30 µM of tubulin is slightly above the $Cc_{MT\ nucleation}$ (*Figure 2C*), we first confirmed that tubulin alone did not form rings or oligomers on ice, and did not grow into microtubules after 10 min of incubation at 37°C in the cryo-EM condition (*Figure 5—figure supplement 3* and *Figure 8—figure supplement 2A, B*).

Consistent with our negative stain EM results of tubulin-GFP-CC1-CKK, many tubulin rings or tubulin oligomers with crescent shapes were observed in all the conditions (*Figure 8A*, red segmentations in *Figure 8B and C*, and *Figure 8—figure supplement 2C*). Short tubulin sheets were also observed on ice (green in *Figure 8B*). Furthermore, we detected semicircular or crescent-shaped tubulin oligomers that were multiply stacked and partly flattened, suggestive of the intermediates between rings and sheets (orange in *Figure 8B*).

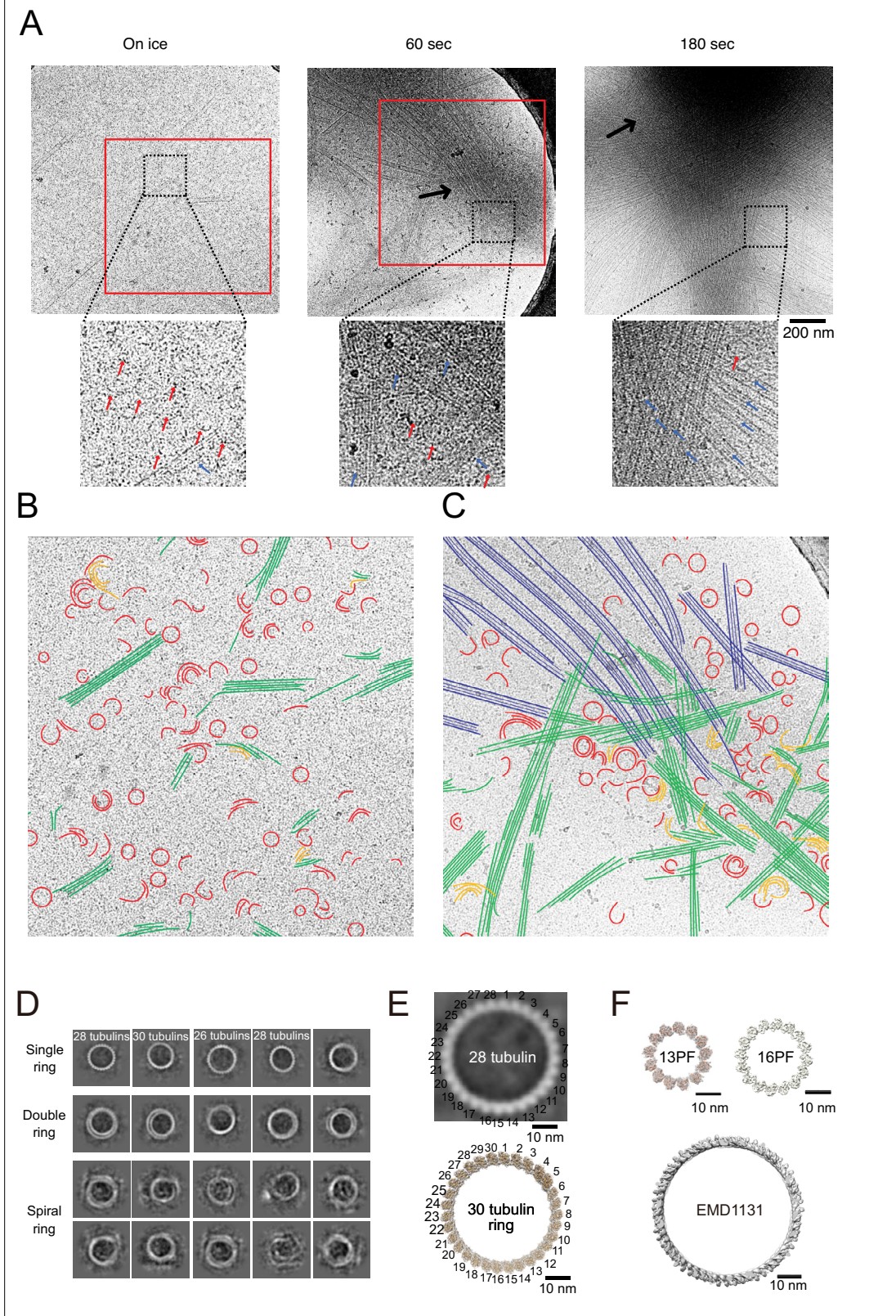

**Figure 8.** Calmodulin-regulated spectrin-associated protein 2 (CAMSAP2) induced microtubule nucleation intermediates visualized by time-lapse cryo-electron microscopy (EM). Representative EM images are shown from at least three independent assays. (**A**) Snapshots of growing microtubule intermediates at different time points. The contrast of the micrographs was adjusted to be slightly higher than the original (see *Figure 8—figure supplement 2C* for raw images). The black arrows indicate high-density condensed areas, red squares are segmented areas. Red arrows in the

*Figure 8 continued on next page*

*Figure 8 continued*

zoomed images indicate tubulin-rings or semi-rings, and blue arrows indicate microtubules or sheets. (**B**) Segmentation of the structural elements of micrographs from panel (**A**) (on ice, red box). Tubulin rings, red; intermediates between ring and sheet, orange; tubulin sheets, green. (**C**) Segmentation of the structural elements of the micrographs from (**A**) (60 s, red box). Tubulin rings, red; intermediates between ring and sheet, orange; tubulin sheets, green; microtubules, blue. (**D**) 2D classification of tubulin rings. Rings of different shapes and sizes were observed, including single rings, spiral rings, and double rings. (**E**) Comparison of 2D average of 28 tubulin rings (top) with the reported tubulin rings produced by the 30 tubulin ring consists of longitudinal contacts (EMD-7026). 30 tubulin ring (EMD-6347) was generated by cropping outer tubulin ring from the microtubule-KLP10A map (EMD-7026) using its model (PDB: 6b0c) as a guide. (**F**) Thirteen and sixteen (EMD-5196) PF microtubule (top) as examples for the ring diameters made by lateral contacts and the tubulin longitudinal tube (EMD-1131) (bottom) as an example for the ring diameter made by longitudinal contacts. Sixteen protofilaments are the thickest microtubule in EMDB. The scale bars indicate 10 nm. The cryo-Electron Tomography (ET) reconstruction of Cam2-asters is available in *Figure 8—figure supplement 1* and *Video 5*.

The online version of this article includes the following figure supplement(s) for figure 8:

**Figure supplement 1.** Cryo-electron tomographic reconstruction of the growing Cam2-asters.

**Figure supplement 2.** Cryo-electron microscopy (Cryo-EM) visualization of Cam2-aster formation.

**Figure supplement 3.** Segmentation of the structural elements of the micrographs at 60 s.

To investigate the ring structures in further detail, we performed 2D class averaging of rings from cryo-EM micrographs (*Figure 8D*). The rings were not uniform; single rings, double rings, and spiral rings were observed. Depending on the lengths of tubulin oligomers, several forms of tubulin rings were produced. 2D classification of the rings clearly showed bumps around the ring surface corresponding to the individual tubulin monomers (*Figure 8E*). The diameters and pitches of the bumps resembled those of longitudinal protofilament rings rather than the tubulin rings made by the lateral contacts like the γ-TuRC (*Figure 8E and F*; *Benoit et al., 2018*; *Sui and Downing, 2010*; *Wang and Nogales, 2005*).

As described above, tubulin rings, possible ring-to-sheet intermediates, and tubulin sheets were uniformly distributed throughout the grid surface of on ice sample (*Figure 8A*). After 1 min of incubation at 37°C, tubulin sheet formation is stimulated. Rings and short sheets tended to gather to form a high-density condensed area (black arrow in *Figure 8A*, 60 s; see also tomogram of growing Cam2-asters formed through 1 min of incubation with CAMSAP2-FL in *Figure 8—figure supplement 1* and *Video 5*). After 3 min of incubation, the condensates became denser, forming the central regions of Cam2-asters (black arrow in *Figure 8A*, 180 s). These observations suggest that incubation at 37°C induces the condensation of tubulin intermediates with CAMSAP2, which stimulates tubulin sheet and microtubule formation, resulting in the rapid growth of Cam2-asters.

We more closely investigated the structural details of the developing aster centres. They were filled with tubulin rings, possible ring-sheet intermediates, sheets, and short microtubules (*Figure 8C* and *Figure 8—figure supplement 3*). Rings were also aligned along the elongating tubulin sheets or growing microtubules that radiated from the condensates, although only a few rings were observed around mature microtubules connecting between condensates. Some rings and sheets assembled together, resembling the process of integration of protofilaments into tubulin sheets (orange in *Figure 8C*). Around the regions where numerous rings existed, the ends of microtubules or tubulin sheets often exhibited partially curved sheet-like structures characteristically observed for growing microtubules (*Figure 8—figure supplement 2D*). We also note that fully curved ends often observed for depolymerizing microtubules were rarely observed.

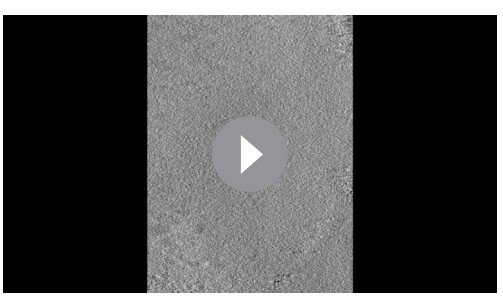

**Video 5.** Cryo-electron tomographic reconstruction of the growing Cam2-aster. SIRT processed cryo-tomographic reconstruction of the growing Cam2-aster produced by incubation of 10 μM tubulin with 1 μM full-length calmodulin-regulated spectrin-associated protein 2 for 1 min at 37°C.

https://elifesciences.org/articles/77365/figures#video5

## Discussion

We reported here that CAMSAP2 functions as a nucleator of microtubule formation from soluble αβ-tubulin dimers through forming condensates, at least in in vitro conditions. CAMSAP2 reduces

the tubulin concentration required for microtubule polymerization from 20 µM to 1–2 µM, the values equivalent to the tubulin $Cc_{MT\ polymerization}$. The acceleration of microtubule nucleation seems to be achieved by the efficient formation of microtubule polymerization intermediates at the inside of the CAMSAP2 condensates, explaining how CAMSAP2 serves as a nucleation centre.

The CASMAP2 forms condensates via LLPS, as it retains liquid-like fluidity and is reversible and sensitive to temperature and salt concentration. Notably, CAMSAP2 could form a condensate at concentrations close to a physiological concentration (*Figure 3D*). When artificial CAMSAP2 foci are made, radial elongation of microtubules from the foci also occurs in its near-physiological concentrations of CAMSAP2 (*Figure 4H*). These results suggest that the LLPS-dependent formation of CAMSAP2 is likely to contribute to the microtubule nucleation not only in vitro but also at the physiological conditions of cells.

Microtubules radiate out from each CAMSAP2 condensate, forming a Cam2-aster. The Cam2-aster resembles the centrosomal aster in which γ-TuRC generates the templated nucleation (*Moritz et al., 1995*; *Roostalu and Surrey, 2017*; *Wieczorek et al., 2020*; *Zheng et al., 1995*). However, the molecular behaviours inside the Cam2-aster are entirely different from those in the centrosomal aster. Within the CAMSAP2 condensates, large numbers of tubulin rings and sheets are produced. Unlike the γ-TuRC in which γ-tubulins contact laterally to make the ring template, the rings observed inside the CAMSAP2 condensates are produced by the longitudinal growth of αβ-tubulin to form a protofilament ring. The fluorescence detection of CAMSAP2 verified its localization at the inside of the condensate. On the other hand, how CAMSAP2 binds to tubulin to produce tubulin rings and sheets remains elusive and is a subject for future study.

The CKK domain of CAMSAPs has been reported to bridge two tubulin dimers in neighbouring protofilaments (*Atherton et al., 2017*). This binding pattern apparently does not support the single ring formation. The cryo-EM structure of the TPX-2-microtubule complex provides us with a possible insight into the mechanism for this process (*Zhang et al., 2017*). TPX-2 uses two elements, ridge and wedge, to stabilize the longitudinal and the lateral contacts of a microtubule. Given that CKK of CAMSAP2 stabilizes the lateral contacts to bridge two tubulin-dimers (*Atherton et al., 2017*) and plenty of CAMSAP2 molecules gather in the condensate, multiple CAMSAP2 molecules could also connect tubulin-dimers longitudinally through multiple CKK domains. The future high-resolution structural study of the CAMSAP2-microtubule complex needs to test this hypothesis.

In the presence of CAMSAP2, the tubulin rings were produced on ice but without growing into microtubules. After warming to 37°C, tubulin sheets and microtubules increase gradually in and around the condensate. In inverse proportion to their increase, the number of rings decreased. Thus, CAMSAP2 is involved in a tubulin ring formation, and tubulins consisting of the rings might be used as a material for microtubule formation. However, we did not capture the scene where the tubulin ring is directly incorporated into sheets or microtubules. On the other hand, we observed many curved sheets with lower curvatures. Hence, microtubule growth in the condensate might be achieved by dynamic fluctuation among the rings, curved sheets, sheets, and microtubules. Tubulin rings could be produced by making longitudinal contacts between tubulin-dimers (*Figure 8E and F*) or by destabilizing the sheet or microtubule. Actually, tubulin rings have long been thought to be microtubule depolymerization products (*Erickson and Stoffler, 1996*). To determine the actual functions of tubulin rings observed in the present study, further high-resolution structural studies of the rings complexed with CAMSAP2 or high-speed imaging of microtubule intermediate formation will be necessary.

We propose a model of spontaneous nucleation mediated by CAMSAP2 based on our data (*Figure 9*). First, CAMSAP2 interacts with tubulins, inducing their longitudinal stack formation until they form protofilament rings. This step can occur on ice. Incubation at 37°C then induces co-condensation of CAMSAP2 and tubulins to stimulate the efficient formation of microtubule nucleation intermediates, forming a nucleation centre. Plenty of possible nucleation intermediates are produced inside the condensates, although how these nucleation intermediates grow into the critical nucleus for microtubule formation is still uncertain (*Erickson and Pantaloni, 1981*). Many microtubules efficiently radiate outward using the nucleation intermediates from the nucleation centre, producing the Cam2-aster. In this stage, CAMSAP2 also decorates the microtubule lattice around the minus-end to induce efficient elongation of microtubules. Microtubules from neighbouring asters then join together and generate a microtubule meshwork among the Cam2-asters, finally building the cytoskeletal framework. In the current structural model, therefore, CAMSAP2 plays a dual role: producing a co-condensate of

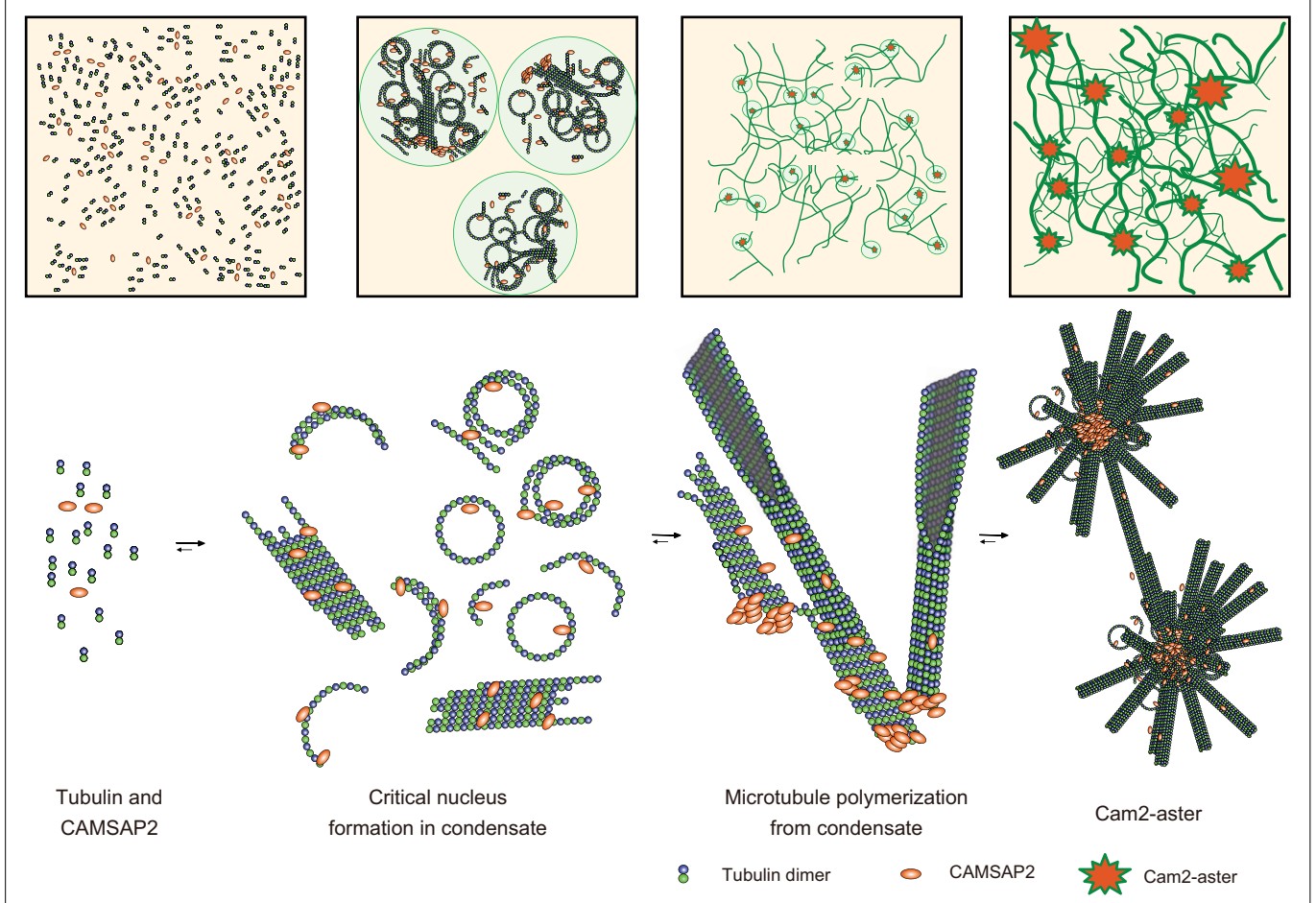

**Figure 9.** Structural model of microtubule nucleation and Cam2-aster formation induced by calmodulin-regulated spectrin-associated protein 2 (CAMSAP2). Structural model of tubulin nucleation, polymerization, and aster formation induced by CAMSAP2, as detailed in the main text. CAMSAP2 shifts the equilibrium to the right, as indicated by the arrow size.

CAMSAP2 and tubulin to function as a microtubule-organizing centre and increasing the efficiency of microtubule polymerization through its binding to the growing microtubule lattice.

Previous studies have indicated that microtubule growth occurs from a focus of Patronin, the *Drosophila* homologue of CAMSAP, which was bound to the cellular cortex via Shot (*Nashchekin et al., 2016*), and in differentiated epithelial cells, microtubules are anchored to the adherens junctions or apical cortices where CAMSAPs are concentrated (*Toya et al., 2016*). Importantly, no γ-TuRC has been detected at these locations. Our in vitro data provide some insight into how CAMSAPs induce microtubule polymerization at the CAMSAP-containing structures. At 50 nM, which is nearly a physiological concentration of CAMSAP proteins in cells (*Liebermeister and Klipp, 2006*; *Wühr et al., 2014*), CAMSAP2 binds to the minus-ends of microtubules but does not induce microtubule polymerization in vitro (*Hendershott and Vale, 2014*; *Jiang et al., 2014*; *Figures 1C and 3*). However, we showed that in the presence of CAMSAP2 condensates fixed to the coverslip, the same concentration of CAMSAP2 could sufficiently induce microtubule polymerization from soluble tubulins in vitro (*Figure 4*). Thus, to support microtubule polymerization from particular structures, both CAMSAP-containing foci and soluble CAMSAP appear to be required.

In vertebrate cells, CAMSAPs generally decorate a single microtubule at its minus-end, not inducing aster formation. Nevertheless, it has been shown that a cluster of Patronin radiates multiple microtubules simultaneously (*Nashchekin et al., 2016*), as observed for the Cam2-asters described here, which suggests that Cam2-aster formation may not be a purely in vitro event. It is possible that excess clustering of CAMSAPs and tubulins is prohibited in vivo through unidentified physiological mechanisms, and CAMSAP-mediated enhancement of microtubule growth, the mechanisms of which

were partly uncovered in this study, likely also works in vivo by forming a smaller or minimal unit of the CAMSAP/tubulin complex. Our findings of the interactions between CAMSAP2 and tubulins reported here using in vitro systems should provide deeper insights into our understanding of their molecular roles in non-centrosomal microtubule network formation in living cells.

## Materials and methods

### Tubulin preparation
Tubulin was purified from porcine brain tissue (*Castoldi and Popov, 2003*). The purified tubulin was labelled with 5 (6)-carboxytetramethylrhodamine succinimidyl ester (TMR) (Santa Cruz Biotechnology, Dallas, TX, USA) or *N*-[6-(biotinamide)hexanoyl]–6-aminohexanoic acid N-succinimidyl ester (Biotin-LC-LC-NHS) (Tokyo Chemical Industry, Tokyo, Japan) as described previously (*Al-Bassam, 2014*).

### Cloning and construct design
The full-length mouse CAMSAP2 gene was PCR-amplified from the CAMSAP2-GFP plasmid (*Toya et al., 2016*) and subcloned between the EcoRI and XhoI restriction sites of the pFastBac1 vector (Thermofisher Scientific). GFP fusion CAMSAP2 full-length with His-Strep tag was generated from PCR-amplify genes of CAMSAP2 gene from CAMSAP2-GFP plasmid (*Toya et al., 2016*), GFP tag gene from CAMSAP2-GFP plasmid (*Toya et al., 2016*), His-Strep2- tag from 438-SNAP-V3 vector a gift from Scott Gradia (Addgene plasmid # 55223) with inserting CATCATCATCATCATCACAGCAGC GGCCTGGTGCCGCGCGGCAGCCAT sequence right in front of Strep II gene by PCR primer, and assembled into PCR amplified pFastBac vector by NEBuilder HiFi DNA assembly (NEB). CAMSAP2 mutant constructs, CC1-CKK, CC3-CKK, and CAMSAP2 CKK were generated by inserting each mouse CAMSAP2 partial PCR fragment flanked by a TEV protease recognition linker (ENLYFQGSSGSSG) and a hepta-His tag at the N-terminus and C-terminus, respectively, into the pGFPS1 vector (*Seki et al., 2008*). This construct contains another metal-binding motif similar to the HAT tag (KDHLIHNHHKHE-HAHA) at the N-terminus of the GFPS1 gene. CMASAP2 ΔCC3 and CAMSAP2 ΔCKK were generated by PCR-amplified whole plasmid DNA without deletion sequence with primer which has complementary region to the opposite end, DpnI digestion, and HiFi DNA assembly reaction. The coding regions of all constructs were verified by Sanger sequencing.

### Protein expression
Full-length CAMSAP2 was expressed in the baculovirus expression vector system using insect cells. Bacmid preparation and V0 and V1 virus production were performed as described previously. The protein was expressed by Sf9 cells after infection with the V1 virus following the titer estimation for quality control protocol (*Imasaki et al., 2018*). Other CAMSAP2 deletion constructs, summarized in *Table 1*, were expressed in *Escherichia coli*. The resulting constructs were transformed into *E. coli* strain KRX (Promega) and cultured in LB medium containing 100 µg/ml ampicillin, 0.05% glucose, and

**Table 1.** List of protein expression constructs used in this study.

| ID | Construct | Amino acids | Vector | Tag (N-terminal) | Tag (C-terminal) |
|---|---|---|---|---|---|
| RN82 | full length | 1–1472 | pFastBac1 | none | 6× His |
| RN63 | CC1-CKK | 696–1,454 | pGFPS1 | HAT*-GFPS1-TEV* | 7× His |
| RN64 | CC3-CKK | 1157–1454 | pGFPS1 | HAT*-GFPS1-TEV* | 7× His |
| RN136 | CKK | 1332–1472 | pGFPS1 | HAT*-GFPS1-TEV* | 7× His |
| RN189 | full length | 1–1472 | pFastBac1 | His_Strep2-3Csite | none |
| RN282 | CC1-CKK-CT | 696–1472 | pET28 | His_Strep2-3Csite | none |
| RN290 | CC3-CKK-CT | 1157–1472 | pET28 | His_Strep2-3Csite | none |
| RN318 | ΔCKK | 1–1331 | pFastBac1 | His_Strep2-3Csite | none |
| RN328 | ΔCC3 | 1–1472 Δ1154–1250 | pFastBac1 | His_Strep2-3Csite | none |

*These tags carry modifications. See the Methods for details.

0.1% rhamnose at 25°C overnight with autoinduction. Each cell suspension was harvested into a 50 ml tube and stored at –80°C.

## Protein purification

For purification of the *E. coli*-expressed constructs, we applied the protocol described below. *E. coli* cells were thawed, resuspended in lysis buffer (40 mM Na-phosphate buffer [pH 7.4], 400 mM NaCl, 5 mM imidazole, 14 mM β-mercaptoethanol, 0.7 μM leupeptin, 2 μM pepstatin A, 1 mM phenylmethylsulfonyl fluoride (PMSF), and 2 mM benzamidine), and lysed by sonication. The lysate was clarified by centrifugation (16,000×*g*), and the supernatant was applied to a HIS-Select Nickel Affinity Gel column (Sigma) equilibrated in lysis buffer. The column was washed with ~10 column volumes of lysis buffer containing 5 mM imidazole. His-tagged proteins were eluted with approximately four column volumes of lysis buffer containing 150 mM imidazole. The elution fractions were pooled and diluted with water to ~150 mM NaCl for cation exchange chromatography. The diluted solutions were clarified by centrifugation using an Avanti JXN30 centrifuge (Beckman Coulter) with a JA30.50 Ti rotor at 20,000×*g* and applied to a HiTrap SP HP column (GE Healthcare; 1 ml) equilibrated in buffer A (20 mM Na-phosphate buffer [pH 7.0], 1 mM Na-ethylenediaminetetraacetic acid (EDTA), and 1 mM dithiothreitol (DTT) containing 150 mM NaCl). The column was washed with the same buffer until a stable baseline was reached. Proteins were eluted with a 20 ml linear gradient from 150 mM to 500 mM at a flow rate of 0.5 ml/min using an ÄKTA Pure instrument (GE Healthcare). The fractions containing the target proteins were pooled and concentrated to less than 0.5 ml using an Amicon Ultra concentrator (Merck Millipore; 10 kDa MWCO). The concentrate was applied to a Superdex 200 Increase 10/300 GL size exclusion chromatography column (GE Healthcare) equilibrated in sizing buffer (20 mM HEPES-KOH pH 7.5, 300 mM NaCl, and 1 mM DTT) at a flow rate of 0.4 ml/min. The peak fractions were pooled, concentrated with an Amicon Ultra 10 kDa MWCO concentrator, flash-frozen in liquid nitrogen, and stored at –80°C until use.

For constructs expressed in insect cells through BEVS, cells were thawed, resuspended in BEVS lysis buffer containing 50 mM HEPES-KOH pH 7.5, 400 mM KCl, 5 mM imidazole, 10% glycerol, 5 mM β-mercaptoethanol, 0.7 μM leupeptin, 2 μM pepstatin A, 1 mM PMSF, and 2 mM benzamidine, and lysed by sonication. The lysate was centrifuged using an Avanti JXN30 (Beckman Coulter) with a JA30.50 Ti rotor at 28,000 rpm (94,830×*g*), and the supernatant was applied to a HIS-select nickel affinity gel column (Sigma) equilibrated in lysis buffer. The column was washed by passing through 20 resin volumes of lysis buffer. The bound proteins were eluted with lysis buffer containing 300 mM imidazole. The eluate was concentrated by an Amicon Ultra 10 kDa MWCO concentrator and applied to a Superose6 Increase 10/300 GL size exclusion chromatography column equilibrated in sizing buffer (50 mM HEPES-KOH pH 7.5, 400 mM KCl, 10% glycerol, and 5 mM DTT). The peak fractions were pooled, concentrated by an Amicon Ultra 10 kDa MWCO concentrator, flash-frozen in liquid nitrogen, and stored at –80°C. The quality of the samples was assessed by SDS-PAGE (*Figure 1B and C*).

## Size exclusion chromatography with multi-angle light scattering

Size exclusion chromatography with multi-angle light scattering (SEC-MALS) experiments were performed at Spring-8 using HPLC system (Agilent 1260 infinity), equipped with a DAWN HELEOS II 8+ (Wyatt Technology) MALS light scattering detector and Optilab rEX differential refractometer (Wyatt Technology). The data was analysed by ASTRA 7 software (Wyatt Technology). About 100 μl of 3 mg/ml CAMSAP2-FL was injected. The Superose6 Increase 10/300 GL column (GE Healthcare) was run with buffer containing 50 mM HEPES (pH 7.5), 400 mM KCl, 10% glycerol, and 2 mM DTT at a flow rate of 0.3 ml/min at 4°C.

## Spontaneous nucleation assay (pelleting assay)

The method was based on that in Wieczorek et al., with some modifications (*Wieczorek et al., 2015*). For the assay testing tubulin alone, the indicated concentrations of tubulin in PEM buffer supplemented with 1 mM GTP (PEM GTP) were incubated on ice for 30 min and then at 37°C for 30 min. Total samples for SDS-PAGE were prepared at this stage. These samples were centrifuged in a TOMY MX-307 for 30 min at 15,000 rpm and 35°C. The supernatant was discarded, and PEM GTP was supplemented for washing, centrifugation, and recovery of the pellet. The pellets were resuspended in cold PEM GTP, incubated on ice for 30 min for depolymerization, and centrifuged

for 30 min at 15,000 rpm and 4°C to remove aggregates or debris. The supernatants were resolved on SDS-PAGE gels, and the bands were quantified by FIJI (ImageJ) (*Schneider et al., 2012*) using a standard made from tubulin aliquots loaded at 1, 2, 3, and 5 µM. For tubulin with CAMSAP2, an assay was performed following the methods of the tubulin-only pelleting assay except that 1 µM CAMSAP2 was added at the initial polymerization step, and CAMSAP2 sizing buffer was used for depolymerization. Since full-length CAMSAP2 is unstable at 37°C without a binding partner such as tubulin, only the CAMSAP2 bound with tubulin was re-solubilized after on-ice incubation and appeared in the SDS-PAGE gel.

## Phase separation

To set buffer condition to be physiological salt concentration, PEM-GTP with 500 mM KCl was mixed with CAMSAP2 and 5× CAMSAP2 was prepared. 5× CAMSAP2 was diluted fivefold with KCl-free PEM-GTP, and immediately loaded into a flow chamber created by sticking slide glass and 18 × 24 mm cover glass with double-sided tape. The flow chamber was placed in the moist chamber with the facing droplet on the cover glass to be down and waited for 20 min, then imaged. Experimental condition was examined using OLYMPUS PROVIS AX80 (OLYMPUS) equipped with the Olympus DP70 (OLYMPUS). Phase separations were imaged by a confocal laser scanning microscope LSM700 (Zeiss). FIJI was used for data analysis (*Schindelin et al., 2012*).

For examining salt concentration, CAMSAP2 concentration was fixed at 1 µM and the KCl concentration was changed from 0 to 500 mM. For examining CAMSAP2 concentration, the KCl concentration was fixed to 100 mM and the CAMSAP2 concentration was changed from 10 to 1000 nM. In both conditions, mixture was loaded into a flow chamber, waited for 20 min, then observed.

For fusion observation, the mixture was prepared in the similar protocol described above with different concentrations of CAMSAP2. CAMSAP2 was set to 4 µM and the salt concentration was set to 100 mM by combining 60 mM NaCl from CAMSAP2 purification buffer and 40 mM KCl for adjustment. In this experiment, observation was started right after CAMSAP2 containing solution was loaded into the flow chamber. Time 00:00 (minutes:seconds) corresponds to the start of the image acquisition.

## Fluorescence recovery after photobleaching

For FRAP, protein solution was prepared same as phase separation experiment. Sample was loaded into flow chamber, the region of interest (ROI) was set to the target droplet, bleached, observed for every 10 s, and measured for 15 min or more. In order to reduce the measurement error due to the drift or movement of the droplet, five set of pictures were combined to one. Each dataset was cropped with ROI in the same size and combined with 'Concatenate', which is a function of FIJI. Next, ROIs were set in the bleached region, with non-bleached region as background, and the average fluorescence intensity was measured. The damage of bleaching due to the beam exposure at the time of data acquisition was corrected and graphed (*Wachsmuth, 2014*).

In the co-localization observation of CAMSAP2 and tubulin, unlabelled tubulin used was mixed with AFDye-labelled tubulin in 9:1 ratio. 5× mixture (5 µM CAMSAP2, 50 µM tubulin) was prepared in salt-free PEM-GTP, diluted fivefold using physiological buffer (PEM-GTP + 100 mM KCl) to be 1 µM CAMSAP2, 10 µM tubulin, 100 mM salt (20 mM NaCl in CAMSAP2 + 80 mM KCl), and loaded into the flow chamber. The chamber was set to the moist chamber, waited for 20 min, then observed.

## Preparation of GMPCPP seeds

TAMRA-tubulin (2.5 µM) and biotin-LC-LC-tubulin (2.5 µM) were polymerized in 100 µl of PEM buffer supplemented with 1 mM Guanosine-5'-[(α,β)-methyleno]triphosphate (GMPCPP) and 1 mM DTT at 37°C for 20 min. Equal amounts of prewarmed PEM buffer were added, and the mixtures were ultra-centrifuged at 80,000 rpm for 5 min at 30°C using a TLA100 rotor and an Optima TLX ultracentrifuge (Beckman). The microtubule pellets were resuspended in PEM supplemented with 1 mM DTT. The GMPCPP seeds were aliquoted into PCR tubes (NIPPON Genetics, Tokyo, Japan), snap-frozen in liquid $N_2$, and stored at –80°C. The GMPCPP seeds were quickly thawed at 37°C immediately before use.

## Binding of CAMSAP2 on dynamic microtubules

Coverslips (#1.5 H thickness, 22 × 22 mm; Thorlabs Japan, Tokyo, Japan) were washed by ultrasonic bath in 1 M HCl (Fujifilm Wako, Tokyo, Japan) for 3 hr, in deionized distilled water for 3 hr and 1 hr, in 70% ethanol for 1 hr, and then in 99.5% ethanol for 1 hr. Flow chambers were prepared at room temperature. The flow chambers were constructed with double-sided tape (Nichiban, Tokyo, Japan). The volume of each flow chamber was approximately 15 µl. The chambers were incubated with 0.5 mg/ml PLL-PEG-biotin (SuSoS, Dübendorf, Switzerland) for 10 min followed by 0.5 mg/ml streptavidin (Fujifilm Wako, Tokyo, Japan) for 2 min. The GMPCPP seeds were incubated for 5 min and washed with 50 µl of assay buffer (100 mM PIPES pH 6.8, 100 mM KCl, 2 mM MgCl$_2$, 1 mM EGTA, 0.5% Pluronic F-127, 0.1 mg/ml Biotin-BSA, 0.2 mg/ml κ-casein). Then, 10 µM tubulin, 0.5 µM TAMRA-tubulin, and 50 nM CAMSAP2 in assay buffer supplemented with 10 nM protocatechuate-3,4-dioxygenase (PCD) (Oriental Yeast, Tokyo, Japan), 2.5 mM protocatechuic acid (PCA) (Fujifilm Wako) and 1 mM Trolox (Fujifilm Wako) were added into the chamber and observed by TIRF microscopy at 37°C. The TIRF system consisted of a Ti2E microscope (Nikon, Tokyo, Japan) equipped with a Ti2-LAPP TIRF system (Nikon), an iXon Life 897 EMCCD camera (Andor), a CFI Apochromat TIRF lens (NA 1.49, x100) (Nikon), and a heating stage (Tokai Hit, Fujinomiya, Japan). Images were obtained every 5 s for 20 min.

## Time-lapse imaging of CAMSAP2-dependent aster formation

PLL-PEG-biotin coating was performed as described above. Instead of biotinylated GMPCPP MTs, a biotinylated anti-GFP antibody (MBL, Tokyo, Japan) diluted 1:10 in PEM buffer supplemented with 100 mM KCl was added into the chamber and incubated for 2 min. The chamber was washed with 50 µl of assay buffer. CAMSAP2-GFP (0.5 µM), which was diluted in the assay buffer to form CAMSAP2 condensates and then the solution, was incubated for 5 min. The chamber was washed with 50 µl of assay buffer again. We checked the binding of CAMSAP2 condensates on the glass surface at this stage. Then, 10 µM tubulin, 0.5 µM TAMRA-tubulin, and the indicated concentrations of CAMSAP2 in assay buffer supplemented with 10 nM PCD, 2.5 mM PCA, and 1 mM Trolox were added into the chamber and were observed by TIRF microscopy as described above. Observation was performed at 37°C. Images were obtained every 5 s for 20 min.

## HeLa cell culture and immunofluorescent staining

The HeLa cells used in this study were obtained from JCRB (cat# JCRB9004 HeLa, lot# 08162018). The identity of the HeLa cell line provided from JCRB has been authenticated by STR profiling and confirmed negative for either bacterial, fungal, or mycoplasma contamination. The cells were maintained according to supplier's instructions. Approximately 5000 HeLa cells were seeded on gelatin-coated coverslips placed in a 24-well culture plate and cultured in DMEM/Ham's F-12 (FUJIFILM Wako Pure Chemical 048–29785) supplemented with 10% FBS, 100 µg/ml streptomycin, and 100 U/ml penicillin. Cells were maintained at 37°C in a 5% CO$_2$, 95% humidified incubator. To depolymerize existing microtubules, 10 µM nocodazole was added to cells cultured for 24 hr and then incubated for 50 min. The cells were washed twice with warmed PBS, the medium was exchanged with medium containing no nocodazole, and cells were allowed to recover for a certain time before fixation. For immunofluorescent staining, cells were fixed in methanol for 10 min at –20°C, washed three times in PBS and blocked in blocking buffer (3% BSA in PBS) for 30 min at room temperature. Cells were then incubated with primary antibodies (for CAMSAP2: Proteintech 17880–1-AP diluted 1:1000, for α-tubulin: 12G10, developed and deposited to the Developmental Studies Hybridoma Bank by J. Frankel and M. Nelson, diluted 1:500) diluted in the blocking buffer overnight at 4°C. The cells were washed three times in PBS and incubated with secondary antibodies (Alexa 488-labelled anti-mouse IgG and Alexa 594-labelled anti-rabbit IgG diluted 1:1000) for 1 hr at room temperature. After being washed three times in PBS, specimens were mounted in Fluoroshield (Abcam). Immunofluorescence was visualized using a confocal laser-scanning microscope (ZEISS LSM 700) with 488 nm and 555 nm lasers.

## Grid preparation and data collection for negative stain electron microscopy

Samples, including tubulin, CAMSAP2, and/or CAMSAP2 mutant samples, were diluted with PEM buffer (100mM PIPES pH 6.8, 1mM MgCl$_2$, 1mM EGTA, and 1mM GTP) and incubated on ice for

30 min before blotting. The co-polymerized samples were mixed, incubated on ice for 30 min, and then transferred to 37°C for 1, 3, 10, or 30 min. After incubation, 4 µl of each sample was applied to a glow-discharged carbon-coated 200-mesh Cu grid (EM Japan). Any excess solution was wicked off with filter paper as soon as possible. The grid was stained with 2% uranyl acetate, the excess stain was blotted with filter paper, and the grid was air-dried. The specimens were observed using a JEM-1400 Plus electron microscope (JEOL) operated at 120 kV and equipped with a JEOL Matataki CMOS camera at nominal magnifications of 600-, 10,000-, 20,000-, and 40,000-fold.

### Grid preparation for cryo-electron microscopy

Next, 30 µM tubulin and 3 µM full-length CAMSAP2 or CC1-CKK in PEM buffer (100 mM PIPES-KOH pH 6.8, 1 mM $MgCl_2$, 1 mM EGTA, and 1 mM GTP) were mixed and incubated on ice for 30 min. For induction of microtubule polymerization, the samples were further incubated at 37°C for 1 min or at 37°C for 3 min. All samples were placed onto glow-discharged R2/2 Quantifoil 300-mesh Cu grids with carbon support film, blotted with filter paper (Whatman No. 1) for 2 s, and plunge-frozen in liquid ethane with a Vitrobot Mark IV (Thermofisher Scientific).

### Cryo-electron microscopy

Data acquisition was performed by using a Glacios Cryo-Transmission Electron Microscope operated at 200 kV with a Falcon III EC Direct Electron Detector (Thermofisher Scientific) at 28,000-fold nominal magnification. A total of 176 micrographs were collected with a total dose of 20 $e^-/Å^2$ at a pixel size of 3.57 Å. For the 2D classification, 5881 particles in total were manually picked from 89 micrographs and subjected to two rounds of reference-free 2D classification in cryoSPARC (*Punjani et al., 2017*).

### Cryo-electron tomography

A Titan Krios 300 kV transmission electron microscope equipped with a Falcon II Direct Electron Detector (Thermofisher Scientific) at nominal magnifications of 14,000-fold was used for 3D structure analysis. A cryo-ET tilt series was recorded with TEM Tomography Data Acquisition software. The dose for one image was calculated as 1.5 $e^-/Å^2$. The tilt series was obtained using a Saxton scheme with steps of 3°C up to ±70°C. Sixty-seven micrographs were taken for one series; the total dose was approximately 100 $e^-/Å^2$ at pixel sizes of 5.9 Å (for growing asters). The tilt series images were reconstructed into 3D tomograms by using the IMOD software package (*Kremer et al., 1996*). The micrographs were aligned by cross-correlation and then aligned by tracking 20 nm gold fiducial beads coated on the grid. The aligned micrographs were reconstructed for visual analysis using IMOD SIRT. CTF correction was not performed. Particles from exemplary classes are displayed in *Figure 6E–I*.

## Acknowledgements

We thank R J McKenney at UC Davis for providing the full-length GFP-CAMSAP2 construct and N Kajimura for helping with the electron microscopy data collection at the Research Center for Ultra-High-Voltage Electron Microscopy (Nanotechnology Open Facilities) at Osaka University. We are also grateful to H Ago for supporting the SEC-MALS experiment at Spring-8 and K Ikegami for his critical reading of the manuscript. We thank K Chin for her research management support. This work was supported by the Nanotechnology Platform of the MEXT, Japan; the RIKEN Pioneering Project 'Dynamic Structural Biology'; and the Platform Project for Supporting Drug Discovery and Life Science Research (Basis for Supporting Innovative Drug Discovery and Life Science Research (BINDS)) from AMED under Grant Number JP18am0101082. We acknowledge support from the Japan Society for the Promotion of Science (KAKENHI; 19K07246 to T I, 25221104 to M T, and 19H03396 and 21H05254 to R N), AMED-CREST from the Japan Agency for Medical Research and Development (JP21gm0810013 to R N and JP21gm1610003 to T I), the Japan Science and Technology Agency/PRESTO (JPMJPR14L2 to T I), Moonshot R&D (JPMJMS2024 to R N), FOREST Program (JPMJFR214K to T I), the Takeda Science Foundation to T I and R N, the Mochida Memorial Foundation for Medical and Pharmaceutical Research to T I and R N, the Uehara Memorial Foundation to R N, Bristol-Myers Squibb to R N, and the Hyogo Science and Technology Association to R N.

## Additional information

### Competing interests

Kazuhiro Aoyama: Kazuhiro Aoyama is affiliated with Thermo Fisher Scientific. The author has no financial interests to declare. The other authors declare that no competing interests exist.

### Funding

| Funder | Grant reference number | Author |
|---|---|---|
| Japan Agency for Medical Research and Development | JP21gm0810013 | Ryo Nitta |
| Japan Science and Technology Agency | JPMJPR14L2 | Tsuyoshi Imasaki |
| Moonshot Research and Development Program | JPMJMS2024 | Ryo Nitta |
| Takeda Science Foundation | | Tsuyoshi Imasaki Ryo Nitta |
| Mochida Memorial Foundation for Medical and Pharmaceutical Research | | Tsuyoshi Imasaki Ryo Nitta |
| Uehara Memorial Foundation | | Ryo Nitta |
| Bristol-Myers Squibb | | Ryo Nitta |
| Hyogo Science and Technology Association | | Ryo Nitta |
| Japan Society for the Promotion of Science | KAKENHI 19K07246 | Tsuyoshi Imasaki |
| Japan Society for the Promotion of Science | KAKENHI 25221104 | Masatoshi Takeichi |
| Japan Society for the Promotion of Science | KAKENHI 19H03396 | Ryo Nitta |
| Japan Society for the Promotion of Science | KAKENHI 21H05254 | Ryo Nitta |
| Japan Agency for Medical Research and Development | JP21gm1610003 | Tsuyoshi Imasaki |
| FOREST Program | JPMJFR214K | Tsuyoshi Imasaki |

The funders had no role in study design, data collection and interpretation, or the decision to submit the work for publication.

### Author contributions

Tsuyoshi Imasaki, Conceptualization, Data curation, Formal analysis, Funding acquisition, Investigation, Methodology, Project administration, Resources, Software, Validation, Visualization, Writing – original draft, Writing – review and editing; Satoshi Kikkawa, Ryo Nitta, Conceptualization, Data curation, Formal analysis, Funding acquisition, Investigation, Methodology, Project administration, Resources, Software, Supervision, Validation, Visualization, Writing – original draft, Writing – review and editing; Shinsuke Niwa, Conceptualization, Data curation, Formal analysis, Funding acquisition, Investigation, Methodology, Resources, Software, Validation, Visualization, Writing – original draft, Writing – review and editing; Yumiko Saijo-Hamano, Hideki Shigematsu, Ayako Sakamoto, Data curation, Formal analysis, Investigation, Methodology; Kazuhiro Aoyama, Kaoru Mitsuoka, Data curation, Investigation, Methodology; Takahiro Shimizu, Shinya Taguchi, Data curation, Formal analysis, Investigation, Methodology, Validation, Visualization; Mari Aoki, Yosuke Yamagishi, Investigation, Methodology; Yuri Tomabechi, Conceptualization, Funding acquisition, Investigation, Methodology, Project

administration, Supervision; Naoki Sakai, Data curation, Formal analysis, Investigation, Methodology, Validation, Visualization, Writing – original draft; Mikako Shirouzu, Conceptualization, Data curation, Formal analysis, Funding acquisition, Investigation, Methodology, Project administration, Supervision; Tomiyoshi Setsu, Yoshiaki Sakihama, Formal analysis, Investigation, Methodology; Eriko Nitta, Data curation, Investigation, Methodology, Project administration, Supervision; Masatoshi Takeichi, Conceptualization, Data curation, Project administration, Supervision, Writing – review and editing

### Author ORCIDs
Tsuyoshi Imasaki ⑩ http://orcid.org/0000-0001-5462-1820
Hideki Shigematsu ⑩ http://orcid.org/0000-0003-3951-8651
Kaoru Mitsuoka ⑩ http://orcid.org/0000-0003-1782-675X
Mikako Shirouzu ⑩ http://orcid.org/0000-0002-7997-2149
Shinya Taguchi ⑩ http://orcid.org/0000-0002-9868-0651
Ryo Nitta ⑩ http://orcid.org/0000-0002-6537-9272

### Decision letter and Author response
Decision letter https://doi.org/10.7554/eLife.77365.sa1
Author response https://doi.org/10.7554/eLife.77365.sa2

## Additional files

### Supplementary files
• Transparent reporting form

### Data availability
All data generated or analysed during this study are included in the manuscript and supporting file.

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
