## [Editor Report]

This work investigates the mechanism by which CAMSAP2 nucleates microtubule asters in vitro, with implications for its role in the cell. The authors show the importance of CAMSAP condensates for aster formation and use electron microscopy to suggest that nucleation proceeds vs stabilization of protofilaments and sheets of tubulin.

---

## [Decision Letter]

**Decision letter after peer review:**

[Editors’ note: the authors submitted for reconsideration following the decision after peer review. What follows is the decision letter after the first round of review.]

Thank you for submitting your work entitled "CAMSAP2 organizes a γ-tubulin-independent microtubule nucleation centre" for consideration by *eLife*. Your article has been reviewed by 3 peer reviewers, one of whom is a member of our Board of Reviewing Editors, and the evaluation has been overseen a Senior Editor. The reviewers have opted to remain anonymous. We are sorry to say that, after consultation with the reviewers, we have decided that your work will not be considered further for publication by *eLife* at this stage.

The reviewers thought the topic of the manuscript is potentially interesting for *eLife* but that currently its impact is limited. The physiological significance is not so clear because rather high concentrations are used here, and the reviewers have identified a number of key issues that would strengthen the story significantly.

*Reviewer #1:*

Using light and electron microscopy imaging, Imasaki et al. provide evidence for a role of CAMSAP2 in microtubule nucleation in vitro. The authors show that CAMSAP2 can lower the critical tubulin concentration required for microtubule polymerization using pelleting assays. These experiments were done with high concentration (1uM) of CAMSAP. The authors then used a light microscopy assay to examine polymerization. At high CAMSAP2 concentrations, foci are formed in the presence of tublin which nucleate microtubule polymerization. The authors then find that artificial clustering of CAMSAP2 can support tubulin nucleation, but only in the presence of free CAMSAP. Importantly the concentration of free CAMSAP2 used in these experiments is close to physiological levels. These findings are of interest because previous reports suggest CAMSAPs are not nucleating factors. The requirement for both clustered and free CAMSAP2 is intriguing although not explained by the manuscript.

The authors then use electron microscopy approaches to investigate the mechanism of CAMSAP nucleation. These experiments were all done under conditions of high CAMSAP2 concentration. They image pre- and early nucleation stages of the CAMSAP2-tubulin mixture in a time-resolved manner and observe that condensates form within minutes of incubation at 37°C. Interestingly, the authors observe ring-like structures, whose frequency decreases with increasing duration of microtubule polymerization. Immunogold-labelling indicates CAMSAP2 is bound at the inside of these rings. Finally, 2D classification of the ring-structures shows different morphologies ranging from single to spiral rings. The authors suggest in their model that the rings are curved, longitudinal protofilaments that promote tubulin nucleation.

The manuscript provides evidence that CAMSAP2 can be involved in microtubule nucleation. It will provide the basis for a number of future experiments. Key outstanding questions include whether the same proposed nucleation mechanism takes place in cells and the mechanism underlying the unusual requirement for both tethered and free CAMSAP2.

Comment:

1) All the structural work is done at high concentrations of CAMSAP2. This leaves open the question of whether the intermediates observed are relevant to the lower concentration experiments performed light microscopy. Is it possible to reproduce the light microscopy experiments on an EM grid? Would the authors expect to see rings of tubulin outside the foci even at lower CAMSAP concentrations?

2) It is also important that the authors discuss why they think both tethered and free CAMSAP2 is required for nucleation

*Reviewer #2:*

The authors identify CAMSAP2 and tubulin combine into 'condensates' in vitro that can serve as nucleation centres for microtubules. They use a combination of TIRF microscopy and electron microscopy to investigate this phenomenon, then propose this as a mechanism of non-centrosomal microtubule organisation and nucleation. The nature and mechanism of non-centrosomal microtubule organisation is an important topic and has impact potential, however it remains unclear whether the phenomena they observe have any relevance to cellular contexts.

The authors demonstrate an in vitro condensation and nucleation process, providing images at both light and electron microscopy scales. The findings would be strengthened by an investigation into the speculated mechanisms behind this process, for example, whether rings indeed act as nucleation intermediates and how CAMSAP2 promotes their formation.

1) CAMSAP binding to microtubules and minus end specificity is highly sensitive to salt concentration. The authors move between 0 and 100mM KCl between EM/spontaneous nucleation assays and TIRF assays respectively- this is expected to change the properties of CAMSAP-microtubule association, making cross-comparison difficult.

2) A concern that should be addressed was whether the 'condensates' observed by EM are resulting from an aggregation process under their specific in vitro conditions that has little relevance in cells. Microtubule binding proteins can often bundle microtubules and small tubulin oligomers in such in vitro preps, even those that do not bundle microtubules in cells.

3) Negative stain EM can produce many artefacts, including large aggregates, particularly when applying stain to high concentrations of background protein. Therefore, it is hard to conclude from the images provided that GFP-CC1-CKK and GFP-CC3-CKK constructs actually produce asters. In a previous study, at physiological concentrations a CC3-CKK construct of CAMSAP2 behaved rather like the CKK only (see Jiang et al., Dev Cell, 2014), so it was surprising that they behaved differently here. TIRF assays rather than EM with these constructs would be more appropriate here.

4) The authors conclude the CKK domain cannot nucleate microtubules or induce aster formation without thorough investigation of different conditions, most importantly higher CKK or tubulin concentrations.

5) Several of the cryo-EM images/videos appeared to have contaminating membranous structures. It is unclear where this contamination is from and whether it is affecting the experiments.

6) It is hard to see the rings in Figure 6C and therefore the relevant nanogold position. Cryo-tomography or high quality cryo-EM images improve this figure.

7) Figure 6 supplement 1 is not sufficient to reassure readers of the quality/occupancy of the nanogold labelling. Characterisation of nanogold labelling such as that provided in Guesdon et al., Nat Cell Biol, 2016, would strengthen the manuscript.

8) In figure 7 they move to use 3 μm CAMSAP2 and 30 μm tubulin. This tubulin concentration is above the nucleation concentration for tubulin alone, so now we are observing both CAMSAP-induced and spontaneous tubulin nucleation processes at once. This makes the results/structures observed hard to interpret.

9) The intra-ring density apparently shown in Figure 7 C is unconvincing- a lot of background is expected at this concentration and the authors trim the rings out so the background cannot be seen properly.

10) There is no indication as to how many times the EM experiments were repeated. It is therefore unclear how reproducible the representative images are.

11) The different constructs used have different lengths of c-terminus after the CKK domain- this could affect the results and is not discussed.

The key additions that would strengthen the impact of this manuscript are;

a) Some good evidence that these condensates/asters have cellular relevance

b) An investigation into the mechanism behind CAMSAP2-driven tubulin condensation and microtubule nucleation. I think a plausible mechanism could be that the CCs are required for oligomerisation, and that oligomerisation can cross-link oligomeric tubulin intermediates. CAMSAP2 oligomeric state is only mentioned in passing and could have been investigated in more detail.

c) Not enough was done to convince the reader that the rings observed are really produced by CAMSAP2 and that they are nucleation centres, particularly as it is very hard to see how a microtubule binding protein that binds across protofilaments and intra-dimer produces rings. Some more convincing images of CAMSAP2 bound to the rings would be a start, or good biochemical evidence that CAMSAP2 is bound to the rings.

*Reviewer #3 (Recommendations for the authors):*

The authors find that the microtubule minus end binding protein CAMSAP2 stimulates the formation of microtubules, tubulin rings and tubulin sheets in vitro. The study extends previously published work that showed that the homolog CAMSAP3 stimulates microtubule nucleation in vitro (Roostalu et al., Cell, 2018). Here the authors investigate this nucleation activity in much more detail using a combination of fluorescence and electron microscopy-based in vitro assays. Main novelties are the observation that CAMSAP2 can co-condense with tubulin and form foci for microtubule outgrowth and that CAMSAP2 stimulates also the formation of tubulin rings and sheets that are interpreted as nucleation intermediates. This study is an initial biochemical characterization of the conditions that can lead to CAMSAP2-tubulin co-condensation and the stimulation of microtubule nucleation in vitro. The study provides interesting information for those interested in the mechanism of microtubule nucleation and raises the question of whether such a CAMSAP2 condensate-mediated nucleation mechanism is physiologically relevant.

Figure 2: Is all pelleted tubulin in the presence of CAMSAP2 pelleted, because it is in the form of microtubules? Or can tubulin precipitate with CAMSAP2 in co-condensates as tubulin rings and sheets? This is particularly important to answer for the lower tubulin concentrations to support the authors' claim that CAMSAP2 stimulates the nucleation of 'microtubules' at concentrations as low as 2 μm tubulin. Microscopy seems to be required to provide a clear answer.

Figure 3. The characterization of co-condensate formation appears preliminary: Does CAMSAP2 also condense by itself (in the absence of tubulin). Are condensates liquid droplets (i.e. can they fuse? how mobile are proteins inside (FRAP)?)? Under which conditions do condensates form (buffer, temperature)? A recent study reported that a C-terminal fragment of CAMSAP3 (similar to the CC1-CKK construct used here) promoted microtubule nucleation (Roostalu et al., Cell, 2018). Although the mechanism of promoting nucleation was not the focus of that work, there was no indication of CAMSAP condensation or formation of asters (in the absence of motors that could form asters in that study), instead microtubules appeared to nucleate individually and were randomly dispersed, even at very high CAMSAP3 concentrations.

Figure 3G. Why do patches of surface-immobilized CAMSAP3 form? Is the biotin functionalization of the surface patterned? For an unpatterned, homogeneously functionalized surface, one would expect a homogenous layer of CAMSAP3.

Figure 3I. Nucleation from patches seems inefficient. Only few microtubules are visible, in apparent contrast to the later observations made by cryo-EM. Please explain why the nucleation efficiency appears to be different in fluorescence microscopy and electron microscopy experiments.

Figure 4. It would be informative to also characterize the CAMSAP2 fragments by fluorescence microscopy, particularly as there seem to be differences between fluorescence and electron microscopy observations.

[Editors’ note: further revisions were suggested prior to acceptance, as described below.]

Thank you for resubmitting your work entitled "CAMSAP2 organizes a *γ*-tubulin-independent microtubule nucleation centre through phase separation" for further consideration by *eLife*. Your revised article has been evaluated by Suzanne Pfeffer (Senior Editor) and a Reviewing Editor.

The manuscript has been improved but there are some remaining issues that need to be addressed, as outlined below:

The reviewers felt your manuscript was potentially suitable for *eLife* but requires both experimental revisions and a shortening/revision of the text to make it clearer.

They felt that parts of your new data is valuable (for example the.FRAP data), but that other parts are not currently of the standard required for *eLife*. Please see the individual reviewer comments below.

Some of the questionable experiments could be removed or conclusions be modified without much damage to the manuscript. However, the surface-immobilisation experiments were critical and need to be revisited.

*Reviewer #1:*

The revised manuscript is much improved and addresses previous reviewer comments concerning different conditions used in different in vitro assays. The use of electron microscopy provides useful insight into the steps underlying the nucleation pathway. I anticipate the study will stimulate a number of future experiments.

I found the description of CAMSAP-2 truncations interesting, but Figure 6 difficult to follow. The authors choose different subsets of constructs for different panels. Please could they show the same constructs for each assay? Also, the text discusses the role of the CKK domain in stimulating MT nucleation (Figure 6- supplement 4), but this is not shown in the summary table in Figure 6A. I think it might be helpful to include this.

As a minor comment the manuscript is well written but quite long. The authors may want to try and shorten it to improve its accessibility. I also spotted a few typographical mistakes (e.g. "Cam2-astrers") suggesting some proofreading would be helpful.

*Reviewer #2:*

The authors have made an effort to address a large number of critical comments which is to be commended. Some of the new data provide useful new information. Now the manuscript is rather long and could be streamlined. However, some of the experiments are still not to the standard one would like to see in *eLife*.

1. It is preliminary to conclude from the gel filtration in Figure 1 that CAMSAP2 is a tetramer or oligomer given that its shape may deviate from spherical. SEC-MALS should be performed.

2. It is still not clear what's happening in the pelleting experiment in Figure 2: aren't condensates loaded with tubulin (or nucleation intermediates) expected to pellet? And if they then dissolve after resuspension of the pellet in the "depolymerization" step they will contribute to the detected signal that is interpreted as 'microtubule'. Similar: If there are indeed some other aggregates, why is it clear that they will (at least not in part) be dissolved? It seems preliminary to conclude from these experiments that "microtubules" were detected, and hence that CAMSAP lowers the cc for "microtubule" nucleation.

3. It's also not clear what happens in the surface immobilization experiment in Figure 4 which is quite critical to support the physiological significance of the study that is mostly performed at extremely high CAMSAP2 concentrations. Why are there CAMSAP2 patches on the surface in the absence of CAMSAP2 in solution. It seems that a biotin-PEG slide is (homogenously) coated with streptavidin and then CAMSAP is immobilized on this surface. This should lead to a homogenous CAMSAP signal in the absence of added CAMSAP in solution, and not to dots. If dots are present in this condition, it may indicate that CAMSAP has not been flowed out completely and that the concentration in solution is higher than thought (or that something else unexpected is happening). Independent of this concern, the experiment seems to indicate that locally increasing the CAMSAP concentration on the surface may promote condensation also at lower CAMSAP concentrations in solution. This might be physiologically relevant if indeed local patches of CAMSAP (whose presence is independent of the presence of microtubules) can be observed in cells.

4. It may also be useful to test if accumulation of tubulin in condensates can be demonstrated with tubulins labelled with fluorophores with different hydrophobicity/charge to exclude an artefact (of "co-condensation") induced by the label.

The main question that this manuscript raises is if the observed effects at high CAMSAP concentrations are physiologically relevant. This largely depends on the interpretation of the surface experiment which should therefore be technically very convincing.

---

## [Author Response]

[Editors’ note: the authors resubmitted a revised version of the paper for consideration. What follows is the authors’ response to the first round of review.]

Reviewer #1:Using light and electron microscopy imaging, Imasaki et al. provide evidence for a role of CAMSAP2 in microtubule nucleation in vitro. The authors show that CAMSAP2 can lower the critical tubulin concentration required for microtubule polymerization using pelleting assays. These experiments were done with high concentration (1uM) of CAMSAP. The authors then used a light microscopy assay to examine polymerization. At high CAMSAP2 concentrations, foci are formed in the presence of tublin which nucleate microtubule polymerization. The authors then find that artificial clustering of CAMSAP2 can support tubulin nucleation, but only in the presence of free CAMSAP. Importantly the concentration of free CAMSAP2 used in these experiments is close to physiological levels. These findings are of interest because previous reports suggest CAMSAPs are not nucleating factors. The requirement for both clustered and free CAMSAP2 is intriguing although not explained by the manuscript.The authors then use electron microscopy approaches to investigate the mechanism of CAMSAP nucleation. These experiments were all done under conditions of high CAMSAP2 concentration. They image pre- and early nucleation stages of the CAMSAP2-tubulin mixture in a time-resolved manner and observe that condensates form within minutes of incubation at 37°C. Interestingly, the authors observe ring-like structures, whose frequency decreases with increasing duration of microtubule polymerization. Immunogold-labelling indicates CAMSAP2 is bound at the inside of these rings. Finally, 2D classification of the ring-structures shows different morphologies ranging from single to spiral rings. The authors suggest in their model that the rings are curved, longitudinal protofilaments that promote tubulin nucleation.The manuscript provides evidence that CAMSAP2 can be involved in microtubule nucleation. It will provide the basis for a number of future experiments. Key outstanding questions include whether the same proposed nucleation mechanism takes place in cells and the mechanism underlying the unusual requirement for both tethered and free CAMSAP2.

First of all, we would like to thank this reviewer for his/her proper understanding of our manuscript and for the comments. We took these criticisms to heart and have substantially revised the manuscript.

Comment:1) All the structural work is done at high concentrations of CAMSAP2. This leaves open the question of whether the intermediates observed are relevant to the lower concentration experiments performed light microscopy. Is it possible to reproduce the light microscopy experiments on an EM grid? Would the authors expect to see rings of tubulin outside the foci even at lower CAMSAP concentrations?

As the reviewer pointed out, the light microscopy experiments showed the aster formation from 250 nM of CAMSAP2 (Figure 4C; Figure 2D-E). We also examined the electron microscopy and pelleting assay experiments at different concentrations of CAMSAP2 in polymerization buffer (PEM) or in physiological buffer (PEM with 100 mM KCl), which is the same buffer we used on the light microscopy experiments. Consistent with the results of the light microscopy experiments, both EM and pelleting assay experiments showed the aster formation from 250 nM (Figure 2D, 2E, Figure 5—figure supplement 2). EM observation at higher magnification further showed microtubules and ring-like oligomers around the condensate. Hence, we could reproduce the light microscopy experiments on an EM grid.

2) It is also important that the authors discuss why they think both tethered and free CAMSAP2 is required for nucleation

CAMSAP2 has two functions. One is to induce the intermediate formation by creating condensates or asters through CC3 and the preceding region, which is a new data integrated into the revised manuscript (new section “CAMSAP2 recruits tubulins through LLPS” on page 10 and new Figure 6B). The other is to create microtubules from intermediates through CKK domains which undergo microtubule stabilization through the binding to the microtubule lattice (Atherton et al., Nat. Commun. 2019, Atherton et al., NSMB, 2017, and Figure 6—figure supplement 4B). “Tethered CAMSAP2” represents the former by increasing the local concentration of the CAMSAP2, and the “free CAMSAP2” represents the latter. We include this discussion in the revised manuscript (p.40, lines 822--835).

Reviewer #2:The authors identify CAMSAP2 and tubulin combine into 'condensates' in vitro that can serve as nucleation centres for microtubules. They use a combination of TIRF microscopy and electron microscopy to investigate this phenomenon, then propose this as a mechanism of non-centrosomal microtubule organisation and nucleation. The nature and mechanism of non-centrosomal microtubule organisation is an important topic and has impact potential, however it remains unclear whether the phenomena they observe have any relevance to cellular contexts.The authors demonstrate an in vitro condensation and nucleation process, providing images at both light and electron microscopy scales. The findings would be strengthened by an investigation into the speculated mechanisms behind this process, for example, whether rings indeed act as nucleation intermediates and how CAMSAP2 promotes their formation.

First of all, we would like to thank this reviewer for his/her comment on the potential impact of our manuscript. We also thank him/her for clarifying several issues of our paper especially for the physiological relevance.

1) CAMSAP binding to microtubules and minus end specificity is highly sensitive to salt concentration. The authors move between 0 and 100mM KCl between EM/spontaneous nucleation assays and TIRF assays respectively- this is expected to change the properties of CAMSAP-microtubule association, making cross-comparison difficult.

Thank you for the constructive comment.

We performed both the spontaneous nucleation assay and the electron microscopy (EM) assay in the same buffer (PEM + 100 mM KCl) as TIRF to allow intercomparison (Figure 2D-E; Figure 5-Supplement 2C). As a result, the nucleation to make the condensate and the radial microtubule formation from the condensate were reproduced. Consistent with the TIRF results (Figure 4C), the efficiency of nucleation and the frequency of microtubule formation were decreased in both assays. We also inserted a text as follows: “Microtubule polymerization was detected from 250 nM CAMSAP2 in PEM, and 500 nM CAMSAP2 in PEM + 100 mM KCl, albeit the polymerization efficiency considerably decreased by increasing the salt concentration (Figure 2D and E).” in page 10 lines 239-241.

2) A concern that should be addressed was whether the 'condensates' observed by EM are resulting from an aggregation process under their specific in vitro conditions that has little relevance in cells. Microtubule binding proteins can often bundle microtubules and small tubulin oligomers in such in vitro preps, even those that do not bundle microtubules in cells.

The properties of condensate made of CAMSAP2 were confirmed by the biochemical assay, the light microscopy (LM) assay, and the EM assay. In biochemistry, we found that the condensate could be pelleted down with tubulin at 37 ºC and re-solubilized by incubating the precipitated fraction on ice (Figure 2B-E). The LM assay confirmed that CAMSAP2 alone was found to produce droplet like condensates even at physiological concentrations. We observed the fusion between two condensates and the fluorescence recovery after photobleaching (FRAP), indicating that condensates have a liquid property (Figure 3). In the EM assay, CAMSAP2 alone was found to produce droplets (Figure 5A), and the early stage of condensate was filled with many rings, sheet-like structures, and short microtubules observed by negative staining (Figure 6—figure supplement 3, Figure 7) and Cryo-EM (Figure 8, Figure 8-supplement 2-3). We did not find aggregates nor a microtubule bundling in the early stage of condensate, as is often seen in MAPs (Figures 7A, 8A, 8—figure supplement 3).

3) Negative stain EM can produce many artefacts, including large aggregates, particularly when applying stain to high concentrations of background protein. Therefore, it is hard to conclude from the images provided that GFP-CC1-CKK and GFP-CC3-CKK constructs actually produce asters. In a previous study, at physiological concentrations a CC3-CKK construct of CAMSAP2 behaved rather like the CKK only (see Jiang et al., Dev Cell, 2014), so it was surprising that they behaved differently here. TIRF assays rather than EM with these constructs would be more appropriate here.

It is the crucial point. As this reviewer suggested, we performed a TIRF assay using the full-length, CC1-CKK, CC3-CKK, and CKK (Figure 6B). As a result, we observed condensate formation and radial microtubule elongation from the condensate in the former three constructs, but not in CKK (Figure 6B-C). The size of the condensate decreased in this order. These results are consistent with the negative stain assay (Figure 6D). In addition, we observed that increasing CKK concentration could induce microtubule polymerization below the critical concentration for microtubule nucleation (Cc_MT nucleation_) as described below (Figure 6—figure supplement 4B).

4) The authors conclude the CKK domain cannot nucleate microtubules or induce aster formation without thorough investigation of different conditions, most importantly higher CKK or tubulin concentrations.

Because increasing tubulin concentration will reach the critical concentration of microtubule nucleation (Cc_MT nucleation_), we examined a higher concentration of CKK (10 µM), which is more than 100 times higher than the physiological concentration. Consequently, CKK still could not induce condensate nor aster formation, but could induce microtubule polymerization below the

Cc_MT nucleation_ (page 28, lines 558-559; Figure 6—figure supplement 4B).

5) Several of the cryo-EM images/videos appeared to have contaminating membranous structures. It is unclear where this contamination is from and whether it is affecting the experiments.

It is unclear where this membranous structure is from. However, it would not affect other experiments since we did not find such vesicles in other tomograms nor the repeated cryo-EM pictures.

In addition, during the revision process, it was no longer necessary to include the tomogram of the mature aster. Thus, we have only included the tomograms of immature asters in Figure 8– Supplement 1 to compare the CC1-CKK induced aster with the full-length induced one.

6) It is hard to see the rings in Figure 6C and therefore the relevant nanogold position. Cryo-tomography or high quality cryo-EM images improve this figure.7) Figure 6 supplement 1 is not sufficient to reassure readers of the quality/occupancy of the nanogold labelling. Characterisation of nanogold labelling such as that provided in Guesdon et al., Nat Cell Biol, 2016, would strengthen the manuscript.

We thank this reviewer for this important suggestion. The gold labelling method presented in the initial manuscript could reproducibly incorporate the gold into the tubulin ring (Author response image 1). However, as this reviewer pointed out, to confirm whether the gold observed in the ring would actually represent CAMSAP2, we should observe CAMSAP2 with the stoichiometrically labelled gold. Therefore, we attempted to label gold specifically via His-tag as described in Guesdon et al. (Guesdon et al., Nat Cell Biol, 2016).

**Author response image 1. sa2fig1:** 

When we attached the gold to the N-terminus via His-tag of CAMSAP2, the gold-his-CAMSAP2 was easily aggregated; thus, we concluded it was difficult to use the strategy for labelling. When the gold was attached to the C-terminus, the CAMSAP2-his-Gold was relatively stable and dispersed when we observed it in negative stain EM. Therefore, we first promoted the polymerization of 10 µM tubulin in the presence of CAMSAP2-his before attaching the gold. Consequently, many rings (red arrows) and sheets (green with white outer frame arrows) were consistently observed 30 seconds later (Author response image 1). Next, gold was attached to the Cterminal His-tag of the CAMSAP2, and the polymerization of 10 µM tubulin was promoted in the same way. As a result, the formation of condensate-like structures with the accumulation of gold was observed (Author response image 1 right panel). The gold bound to the microtubule lattice was infrequently observed (Author response image 1, the left panel). However, the accumulation of CAMSAP2-his-gold did not progress to the formation of tubulin-rings, sheets, or microtubules. It is probably due to the steric hindrance caused by gold, which inhibits the microtubule lattice formation. The result was the same with cryo-EM.

Therefore, we decided to withdraw the gold experiment from this paper. Instead, we have added the other important topic, the condensate (LLPS) formation (new section “CAMSAP2 recruits tubulins through LLPS” page 10~), to clarify the dual roles of CAMSAP2, the condensate formation with tubulins and the microtubule stabilization. A discussion about the CAMSAP2 location has been limited to what we knew from the fluorescence experiment, i.e., CAMSAP2 accumulates in the condensate, or it binds to the microtubules lattice during microtubule elongation.

8) In figure 7 they move to use 3 μm CAMSAP2 and 30 μm tubulin. This tubulin concentration is above the nucleation concentration for tubulin alone, so now we are observing both CAMSAP-induced and spontaneous tubulin nucleation processes at once. This makes the results/structures observed hard to interpret.

Sample preparation for Cryo-EM usually requires about ten times higher concentration than negative staining. Therefore, we examined the series of tubulin concentrations to minimize the spontaneous polymerization from tubulin and also allow the observation by Cryo-EM, finally setting the concentration of tubulin at 30 µM and CAMSAP2 at 3 µM. In the revised manuscript, we have revised texts with new figures to confirm that no polymerized microtubule was observed under tubulin-only conditions (p.19, lines 436-443; Figure 5—figure supplement 3; p.30, lines 615619; Figure 8—figure supplement 2B). We also checked that the cryo-EM results were consistent with TIRF and negative staining.

9) The intra-ring density apparently shown in Figure 7 C is unconvincing- a lot of background is expected at this concentration and the authors trim the rings out so the background cannot be seen properly.

As mentioned above about the gold experiment, the localization of CAMSAP2 observed by electron microscopy is still unclear. Thus, the text referring to the intra-ring density were removed from the revised manuscript.

10) There is no indication as to how many times the EM experiments were repeated. It is therefore unclear how reproducible the representative images are.

All the EM experiments shown in the manuscript have been repeated at least three times and can be reproduced at any time. We have added the following statement in each figure legend in the revised manuscript. "Representative EM images are shown from at least three independent assays".

11) The different constructs used have different lengths of c-terminus after the CKK domain- this could affect the results and is not discussed.

We used the former constructs CC1-CKK and CC3-CKK because they are functional and more stable than CC1-CKK-CT and CC3-CKK-CT. In the revised manuscript, we also added new Figure 6—figure supplement 2 to show functional equivalence between these constructs. We have added this discussion on page 21, lines 466-469.

The key additions that would strengthen the impact of this manuscript are;a) Some good evidence that these condensates/asters have cellular relevance

We are very interested in the cellular relevance, albeit it is beyond the scope of this manuscript. To clarify our focus, we wrote in the abstract as follows: “Taken together, these in vitro findings suggest that CAMSAP2 supports microtubule nucleation and growth by organizing a nucleation centre as well as by stabilizing microtubule intermediates.”. It should be examined in the near future to elucidate the cellular mechanism of CAMSAP2.

b) An investigation into the mechanism behind CAMSAP2-driven tubulin condensation and microtubule nucleation. I think a plausible mechanism could be that the CCs are required for oligomerisation, and that oligomerisation can cross-link oligomeric tubulin intermediates. CAMSAP2 oligomeric state is only mentioned in passing and could have been investigated in more detail.

Thank you for pointing out. As this reviewer suspected, our experiments also suggested coiled coil domains are an important region for the condensate and aster formation, which indirectly indicates the oligomerization play important role for the CAMSAP function. As described above, we added the discussion about the possibility that oligomeric CAMSAP2 could connect two tubulin-dimers to proceed with the tubulin nucleation in the revised manuscript (p.38, lines 766770).

c) Not enough was done to convince the reader that the rings observed are really produced by CAMSAP2 and that they are nucleation centres, particularly as it is very hard to see how a microtubule binding protein that binds across protofilaments and intra-dimer produces rings. Some more convincing images of CAMSAP2 bound to the rings would be a start, or good biochemical evidence that CAMSAP2 is bound to the rings.

As this reviewer pointed out, the best is to observe the CAMSAP2 bound to the ring in high resolution or show good biochemical evidence, albeit it spends a lot more time and work (now ongoing in our lab). Instead, we have added new data in the revised manuscript related to the CAMSAP2 condensation, which is essential for the Cam2-aster formation. From the condensation observation by EM, LM, and biochemical analyses, we found a strong correlation between the presence of CAMSAP2 and the generation of tubulin-rings and sheets in the condensate. However, the origin or role of rings is still uncertain in the current situation, and we should perform highspeed imaging and/or solve the high-resolution structures to elucidate what the ring is. This discussion has been included in the revised manuscript (p.38, lines 773-788).

Reviewer #3:The authors find that the microtubule minus end binding protein CAMSAP2 stimulates the formation of microtubules, tubulin rings and tubulin sheets in vitro. The study extends previously published work that showed that the homolog CAMSAP3 stimulates microtubule nucleation in vitro (Roostalu et al., Cell, 2018). Here the authors investigate this nucleation activity in much more detail using a combination of fluorescence and electron microscopy-based in vitro assays. Main novelties are the observation that CAMSAP2 can co-condense with tubulin and form foci for microtubule outgrowth and that CAMSAP2 stimulates also the formation of tubulin rings and sheets that are interpreted as nucleation intermediates. This study is an initial biochemical characterization of the conditions that can lead to CAMSAP2-tubulin co-condensation and the stimulation of microtubule nucleation in vitro. The study provides interesting information for those interested in the mechanism of microtubule nucleation and raises the question of whether such a CAMSAP2 condensate-mediated nucleation mechanism is physiologically relevant.

First of all, we would like to thank this reviewer for his/her proper understanding of our manuscript and for the comments. We took these criticisms to heart and have substantially revised the manuscript.

Figure 2: Is all pelleted tubulin in the presence of CAMSAP2 pelleted, because it is in the form of microtubules? Or can tubulin precipitate with CAMSAP2 in co-condensates as tubulin rings and sheets? This is particularly important to answer for the lower tubulin concentrations to support the authors' claim that CAMSAP2 stimulates the nucleation of 'microtubules' at concentrations as low as 2 μm tubulin. Microscopy seems to be required to provide a clear answer.

We performed a microtubule polymerization from 2 µM tubulin with 1 µM CAMAP2, which is close to the minimum concentration on the pelleting assay, and observed by negative stain EM (Figure 5—figure supplement 2A**)**. Consequently, we could observe the Cam2-aster formation consisting of plenty of microtubules. Hence, we can conclude that CAMSAP2 stimulates the nucleation of 'microtubules' at concentrations as low as 2 μm tubulin.

Figure 3. The characterization of co-condensate formation appears preliminary: Does CAMSAP2 also condense by itself (in the absence of tubulin). Are condensates liquid droplets (i.e. can they fuse? how mobile are proteins inside (FRAP)?)? Under which conditions do condensates form (buffer, temperature)?

We sincerely appreciate this review’s point, which significantly strengthens our manuscript. We performed a series of experiments to characterize the condensate formation. As this reviewer expected, CAMSAP2 itself could form condensate through LLPS. CAMSAP2 condensates fuse and show liquid fluidity by FRAP (new Figure 3E and F). We have added a section “CAMSAP2 recruits tubulins through LLPS” to the manuscript from page 10.

A recent study reported that a C-terminal fragment of CAMSAP3 (similar to the CC1-CKK construct used here) promoted microtubule nucleation (Roostalu et al., Cell, 2018). Although the mechanism of promoting nucleation was not the focus of that work, there was no indication of CAMSAP condensation or formation of asters (in the absence of motors that could form asters in that study), instead microtubules appeared to nucleate individually and were randomly dispersed, even at very high CAMSAP3 concentrations.

We cloned, expressed, and purified the full length CAMSAP3 and tested the ability of aster formation (Figure 5—figure supplement 1). Consequently, CAMSAP3 exhibits the ability to promote the microtubule aster formation, although the microtubule nucleation and growth seemed less efficient than CAMSAP2 (page. 18, lines 396-401).

Figure 3G. Why do patches of surface-immobilized CAMSAP3 form? Is the biotin functionalization of the surface patterned? For an unpatterned, homogeneously functionalized surface, one would expect a homogenous layer of CAMSAP3.

We sometimes see this phenomenon. We use an unpatterned homogeneous surface. The striped pattern might be created by the physical flow because PLL-PEG-biotin, anti-GFP antibody and GFP-CAMSAP are flowed into the flow chamber. We think this is not related to the property of CAMSAP2.

Figure 3I. Nucleation from patches seems inefficient. Only few microtubules are visible, in apparent contrast to the later observations made by cryo-EM. Please explain why the nucleation efficiency appears to be different in fluorescence microscopy and electron microscopy experiments.

We performed the TIRF experiment in the physiological buffer consisting of PEM + 100 mM KCl. Therefore, we performed a pelleting assay (new Figure 2E) and EM analysis using the same buffer for comparison (Figure 5—figure supplement 2). As a result, both experiments consistently showed that Cam2-aster formation with the physiological buffer was inefficient compared to the PEM. It is why Figure 3I in the old manuscript (Figure 4I in the revised manuscript) showed inefficient nucleation, as this reviewer pointed out. We include these results in the revised manuscript (page 10, the paragraph starts from line 236 and page 19 lines 428-434).

Figure 4. It would be informative to also characterize the CAMSAP2 fragments by fluorescence microscopy, particularly as there seem to be differences between fluorescence and electron microscopy observations.

We investigated the aster formation of the CAMSAP2 deletion constructs by TIRF assay (new Figure 6C). These results were consistent with the negative stain EM results (new Figure 6D).

[Editors’ note: what follows is the authors’ response to the second round of review.]

The manuscript has been improved but there are some remaining issues that need to be addressed, as outlined below:Reviewer #1:The revised manuscript is much improved and addresses previous reviewer comments concerning different conditions used in different in vitro assays. The use of electron microscopy provides useful insight into the steps underlying the nucleation pathway. I anticipate the study will stimulate a number of future experiments.

We thank this reviewer for the kind evaluation of our manuscript.

I found the description of CAMSAP-2 truncations interesting, but Figure 6 difficult to follow. The authors choose different subsets of constructs for different panels. Please could they show the same constructs for each assay? Also, the text discusses the role of the CKK domain in stimulating MT nucleation (Figure 6- supplement 4), but this is not shown in the summary table in Figure 6A. I think it might be helpful to include this.

We apologize for the poor quality of our presentation. Along with this reviewer’s suggestion, the names of all constructs were written in a unified manner for clarity. Also, the CKK domain has been included in Figure 6A in the revised manuscript.

Reviewer #2:The authors have made an effort to address a large number of critical comments which is to be commended. Some of the new data provide useful new information. Now the manuscript is rather long and could be streamlined. However, some of the experiments are still not to the standard one would like to see in eLife.1. It is preliminary to conclude from the gel filtration in Figure 1 that CAMSAP2 is a tetramer or oligomer given that its shape may deviate from spherical. SEC-MALS should be performed.

We thank this reviewer for suggesting an essential experiment for protein quality evaluation. We performed an SEC-MALS experiment with an HPLC system (Agilent 1260 infinity), equipped with a DAWN HELEOS II 8+ (Wyatt Technology) MALS light scattering detector and Optilab rEX differential refractometer (Wyatt Technology) (also written in Materials and methods, Size-exclusion chromatography with multi-angle light scattering (SEC-MALS)). The sample was concentrated to the maximum concentration of the CAMSAP2-FL of 3 mg/ml, which could not be higher than this concentration. It is typical behavior for the large multi-domain proteins containing a long IDP region. A hundred microliters of protein, which is the best volume to get the high and sharp signal for the column in our experience, were applied to the Superose6 Increase 10/300 GL column and analysed. As a result, CAMSAP2-FL showed a size of 122 ± 1.1 kDa (Figure 1—figure supplement 1) and we judged CAMSAP2-FL as a monomer. We have also included a short discussion describing the possibility that plenty of CAMSAP2 monomers that gather in the condensate could connect two tubulin-dimers longitudinally through multiple CKK domains (pp. 21, lines 518-525).

2. It is still not clear what's happening in the pelleting experiment in Figure 2: aren't condensates loaded with tubulin (or nucleation intermediates) expected to pellet? And if they then dissolve after resuspension of the pellet in the "depolymerization" step they will contribute to the detected signal that is interpreted as 'microtubule'. Similar: If there are indeed some other aggregates, why is it clear that they will (at least not in part) be dissolved? It seems preliminary to conclude from these experiments that "microtubules" were detected, and hence that CAMSAP lowers the cc for "microtubule" nucleation.

We appreciate this suggestion. We agree with the point, and we re-wrote our manuscript carefully using the word 'microtubule' in this section.

3. It's also not clear what happens in the surface immobilization experiment in Figure 4 which is quite critical to support the physiological significance of the study that is mostly performed at extremely high CAMSAP2 concentrations. Why are there CAMSAP2 patches on the surface in the absence of CAMSAP2 in solution. It seems that a biotin-PEG slide is (homogenously) coated with streptavidin and then CAMSAP is immobilized on this surface. This should lead to a homogenous CAMSAP signal in the absence of added CAMSAP in solution, and not to dots. If dots are present in this condition, it may indicate that CAMSAP has not been flowed out completely and that the concentration in solution is higher than thought (or that something else unexpected is happening).

We are sorry for the lack of experimental detail. Experimental detail is following (Figure 4D, 4F).

The coverslip was coated with PLL-PEG-Biotin, followed by the streptavidin coat. A biotin-labeled anti-GFP antibody was then incubated and fixed on the glass. After fixation, the excesses antibody was washed out, and 1 μm GFP-CAMSAP2-FL diluted in PEM supplemented with 100 mM KCl was loaded on the slide. After the incubation, GFP-CAMSAP2-FL solution was completely washed out. And finally, an observation buffer containing 10 μm tubulin and indicated concentrations of GFP-CAMSAP2-FL (0 – 1,000 nM) were loaded.

As this reviewer described, the streptavidin was homogeneously coated, shown by the GFP alone homogeneously distributed on the glass surface (Author response image 2). In the case of CAMSAP2, however, 1 μm GFP-CAMSAP2-FL forms condensates in PEM–100 mM KCl solution (Figure 3B-D). Hence, GFP-CAMSAP2-FL forms condensates in solution before attaching to the glass surface through GFP antibodies. In other words, GFP-CAMSAP2-FL binds to the glass surface as a condensate so that we can see dotted signals of GFP-CAMSAP2-FL on the glass surface even in the absence of added CAMSAP in the solution. In addition, GFP-CAMSAP2-FL condensates on the glass surface are stable in the absence of free GFP-CAMSAP2-FL in solution for at least 30 min.

**Author response image 2. sa2fig2:** GFP attached on the glass surface represents homogeneous signals, unlike GFP-CAMSAP2-FL.

We added this explanation to avoid misleading readers (pp. 10, lines 245-254) and inserted detailed procedures to the method in the revised manuscript (pp. 44, lines 1299-1310).

Independent of this concern, the experiment seems to indicate that locally increasing the CAMSAP concentration on the surface may promote condensation also at lower CAMSAP concentrations in solution. This might be physiologically relevant if indeed local patches of CAMSAP (whose presence is independent of the presence of microtubules) can be observed in cells.

We sincerely appreciate this suggestion. As observed in the new Figure4—figure supplement 1, we could detect the intrinsic CAMSAP patches in HeLa cells during the microtubule network formation (0-3 minutes incubation at 37 ºC after the nocodazole treatment). This result supports the physiological relevance of CAMSAP condensate in cells.

4. It may also be useful to test if accumulation of tubulin in condensates can be demonstrated with tubulins labelled with fluorophores with different hydrophobicity/charge to exclude an artefact (of "co-condensation") induced by the label.The main question that this manuscript raises is if the observed effects at high CAMSAP concentrations are physiologically relevant. This largely depends on the interpretation of the surface experiment which should therefore be technically very convincing.

Thank you for this comment. We have examined both AFDye 594 (Figure 3) and tetramethylrhodamine (TMR) (Figure 4). It confirms that tubulin accumulation in condensates can be observed with tubulins labeled with fluorophores with different hydrophobicity/charges.

A previous study has shown that TMR has the lowest artefacts among the red fluorescent probes (Zanetti-Domingues et al., Plos One, 2013). A lot of studies have also used TMR-labelled tubulin for TIRF experiments. Therefore, we used TMR for the experiments in Figure 4 to achieve the high S/N with the lowest artefacts.